# WikiChat: Stopping the Hallucination of Large Language Model Chatbots by Few-Shot Grounding on Wikipedia

**Sina J. Semnani**  **Violet Z. Yao**[*]  **Heidi C. Zhang**[*]  **Monica S. Lam**

Computer Science Department
Stanford University
Stanford, CA
`{sinaj, vyao, chenyuz, lam}@cs.stanford.edu`

## Abstract

This paper presents the first few-shot LLM-based chatbot that almost never hallucinates and has high conversationality and low latency. WikiChat is grounded on the English Wikipedia, the largest curated free-text corpus.

WikiChat generates a response from an LLM, retains only the grounded facts, and combines them with additional information it retrieves from the corpus to form factual and engaging responses. We distill WikiChat based on GPT-4 into a 7B-parameter LLaMA model with minimal loss of quality, to significantly improve its latency, cost and privacy, and facilitate research and deployment[1].

Using a novel hybrid human-and-LLM evaluation methodology, we show that our best system achieves 97.3% factual accuracy in simulated conversations. It significantly outperforms all retrieval-based and LLM-based baselines, and by 3.9%, 38.6% and 51.0% on head, tail and recent knowledge compared to GPT-4. Compared to previous state-of-the-art retrieval-based chatbots, WikiChat is also significantly more informative and engaging, just like an LLM.

WikiChat achieves 97.9% factual accuracy in conversations with *human* users about recent topics, 55.0% better than GPT-4, while receiving significantly higher user ratings and more favorable comments.

## 1 Introduction

Recent dramatic advances in LLM chatbots have made them indispensable tools for millions of people (Hu, 2023) who have come to rely on their broad skill set. Yet, LLM chatbots are prone to providing misleading information, or *hallucination* (Bang et al., 2023), often using a convincing and confident language. Notably, LLMs do not speak accurately about *recent* events that occurred

---

after their pre-training, and are far less knowledgeable about less popular, or *tail*, topics (Mallen et al., 2022; Sun et al., 2023). Therefore, for knowledge-intensive tasks (Lewis et al., 2020), users need to painstakingly verify any information they receive with external sources lest they be misled.

This paper focuses on three metrics for knowledge-intensive dialogues: factuality, conversationality, and latency. A knowledge-based chatbot needs to be first and foremost *factual*. We assume access to a source of trusted text corpus; here the English Wikipedia is assumed to be factual. While LLMs tend to hallucinate, they can carry out natural and engaging conversations rather than giving dry answers to users' questions. We refer to the ability to give relevant, informational, natural, non-repetitive and temporally accurate answers collectively as *conversationality*. We single out *latency* as the third metric of focus because solutions addressing factuality like Gao et al. (2023); Jiang et al. (2023); Trivedi et al. (2023); Zhao et al. (2023) tend to incur a high latency that degrades user experience and hinders adoption.

### 1.1 Current Approaches

The basis of factuality in this work is information retrieval (IR), which bases the chatbot's responses on retrieved information from a trusted corpus (Zhang et al., 2023; Li et al., 2022; Shuster et al., 2021). The *retrieve-then-generate* approach generates a response from data retrieved with the user query (Lewis et al., 2020; Trivedi et al., 2023; Izacard et al., 2022; Shuster et al., 2022b; Chen et al., 2021). Previous work is either not evaluated on conversational tasks (Lewis et al., 2020; Trivedi et al., 2023; Shi et al., 2023), or as we show in this paper, is likely to generate irrelevant and unnatural outputs when properly evaluated (Izacard et al., 2022). More importantly, chatbots based on retrieve-then-generate pipelines may still hallucinate. In popular academic datasets like Wizard of

---

[*]Equal contribution

[1]Code and model available at `https://github.com/stanford-oval/WikiChat`.

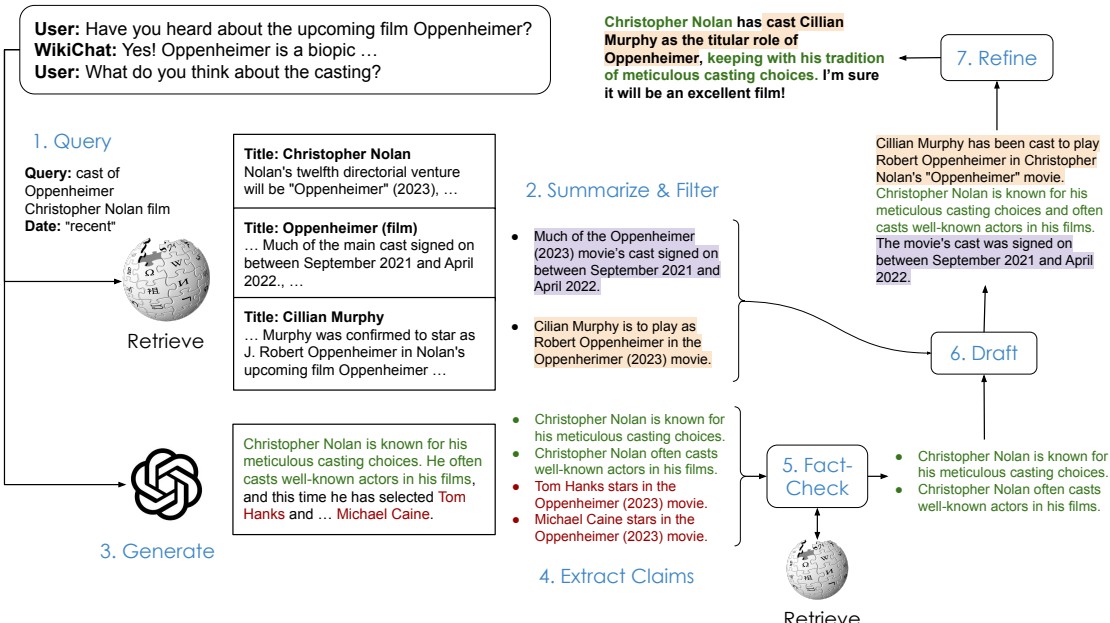

Figure 1: All WikiChat components, and a sample conversation about an upcoming movie, edited for brevity. The steps taken to generate a response include (1) generating a query to retrieve from Wikipedia, (2) summarizing and filtering the retrieved passages, (3) generating a response from an LLM, (4) extracting claims from the LLM response (5) fact-checking the claims in the LLM response using retrieved evidence, (6) drafting a response, and (7) refining the response.

Wikipedia (Dinan et al., 2019) and Wizard of Internet (Komeili et al., 2022), which are widely used to train retrieve-then-generate chatbots, crowdworkers are free to add ungrounded information to their responses; in a GPT-4-based commercial system like Bing Chat, only 58.7% of the facts generated are grounded in what it retrieves (Liu et al., 2023a).

Another IR approach is to fact-check a system's outputs and remove errors (Gao et al., 2023; Jiang et al., 2023). When applied to LLM chatbots, the responses are conversational, but as shown in this paper, are lacking in content for recent or tail topics. Other IR approaches require expensive changes to the pre-training process of language models (Lewis et al., 2020; Guu et al., 2020).

Complementary to IR, *Knowledge Editing* updates model weights to include recent knowledge as it becomes available (De Cao et al., 2021; Meng et al., 2022; Mitchell et al., 2022). Similarly, *Continual Learning* can be used to add new knowledge to LLMs (Jang et al., 2022). While these approaches improve factuality on recent knowledge, they cannot address the tail knowledge problem.

## 1.2 WikiChat Overview

This paper presents WikiChat, the first few-shot chatbot that provides up-to-date and fact-checked information with high conversationality and low latency.

**Few-Shot Knowledge-Grounded Chatbots.** Our 7-stage pipeline (Figure 1) combines the best of IR approaches: We (1) use the user utterance to retrieve information that LLMs may not be aware of, and (2) leverage the generative power of LLMs by asking them for responses and fact-checking them. All this curated information is used to draft and refine the final response.

It is not easy to stop LLMs from hallucinating. In retrieve-then-generate pipelines, when IR does not retrieve any relevant information or when no relevant information is available in the knowledge corpus, LLMs hallucinate to pick up the slack. Thus, WikiChat summarizes and filters the retrieved information instead of generating a response directly. We fact-check every claim generated by LLMs separately and teach the system to say "I don't know" when necessary. We teach it to understand the time context; e.g. a future tense in an article may refer to a past event at the time of the conversation. Most importantly, we do not prematurely optimize for speed by forgoing these needed steps, but rely on model distillation to reduce the latency only once high quality is achieved.

The resulting pipeline is not specific to any corpus. While this paper applies this pipeline to

Wikipedia, the largest corpus of curated knowledge, to create WikiChat, it is applicable to any free-text corpus, including personal and corporate confidential information. The pipeline is not specific to any LLM either, and we apply it to three different LLMs in this paper.

**Distillation for improved latency, affordability, and privacy**. Not only is our 7-stage LLM-based pipeline slow and expensive, sending user data to LLM APIs does not provide the privacy and confidentiality needed by many applications. The simplicity in each of our stages makes it possible to effectively distill our best system into a smaller multi-tasking model, which is responsive and affordable, and can be deployed locally for privacy. We release this model to aid further research and reproducibility of our results.

**Evaluation of LLM-based agents**. We find that LLM-based chatbots have surpassed the quality of systems that conventional static crowdsourced benchmarks (Dinan et al., 2019; Komeili et al., 2022) were meant to evaluate. For example, these benchmarks mainly evaluate the ability to chat about the *head* knowledge, which LLM chatbots are already very good at. We devise a human-and-LLM hybrid evaluation methodology that can adequately analyze all chatbots, regardless of whether they are knowledge-grounded or LLM-based.

## 1.3 Contributions

We create **a factual and engaging open-domain chatbot** with a 7-stage pipeline using a few-shot prompted LLM, as shown in Figure 1. We validate the concept with three GPT-4, GPT-3.5, and LLaMA (Touvron et al., 2023) based chatbots grounded in Wikipedia.

Our experiments with simulated users show that the GPT-4-based WikiChat (WikiChat $_{G4}$) achieves a 97.3% factual accuracy of its claims in simulated conversations. Each version of WikiChat is more accurate than the LLM it is based on by an average of 31.2%, 27.8% and 50.2% for GPT-4, GPT-3.5 and LLaMA respectively. It also outperforms the fine-tuned SOTA retrieval-based Atlas (Izacard et al., 2022) in factuality and, unlike Atlas, matches the conversationality of LLMs.

Our *real* user study shows that WikiChat achieves 97.9% in factuality on conversations of recent topics, 2.3 times more factual than GPT-4, while receiving higher user ratings.

We are the first to demonstrate the **feasibility of distilling a multi-part system built with in-context learning (Brown et al., 2020) into a smaller yet effective model**. Our distillation of WikiChat $_{G4}$ into a 7B-parameter LLaMA achieves a factual accuracy of 91.1%, outperforming much larger baselines, while having 3.2 times lower end-to-end latency than its teacher model.

We introduce an **efficient and effective human-and-LLM methodology for evaluating knowledge-grounded chatbots** in settings beyond what is possible with just crowdsourcing.

## 2 Related Work

**Knowledge-Grounded Chatbots.** Information retrieval is commonly used to develop knowledge-grounded chatbots (Shuster et al., 2021). Blender-Bot 2 (Chen et al., 2021) incorporates Internet search. SeeKeR (Shuster et al., 2022a) outperforms BlenderBot 2 (Chen et al., 2021) by utilizing a single language model for three modular tasks: generating search queries, producing relevant knowledge from retrieved documents, and generating the final response. BlenderBot 3 (Shuster et al., 2022b) fine-tunes a 175B-parameter OPT (Zhang et al., 2022) on the combination of 20 question answering and dialogue datasets. Atlas (Izacard et al., 2022) is a state-of-the-art model on the KILT benchmark (Petroni et al., 2021), which consists of 11 knowledge-oriented tasks including Wizard of Wikipedia (Dinan et al., 2019).

**Evaluating Factuality.** FEVER (Thorne et al., 2018) is a popular crowdsourced dataset that compares claims against evidence retrieved from Wikipedia, and was extended to dialogues by Gupta et al. (2022). The state-of-the-art system on this dataset (Krishna et al., 2022) has an accuracy of 81% when compared against human labels. $Q^2$ (Honovich et al., 2021) uses question answering and natural language inference models to evaluate the factuality of dialogue agents. Dziri et al. (2022) compare this and several other automatic metrics to human evaluation and find that automatic metrics fall significantly behind human performance, and rely on spurious correlations. Concurrently and similarly to our work, Min et al. (2023) break down long system outputs to claims and evaluate their factuality using retrieval. Their best model has a significant 13% error rate on individual claims. Given the high error rates for these automatic metrics, our evaluation methodology (Section 5) incorporates human judgement where needed. Approaches

like Qian et al. (2023) and TripleScore (Goodrich et al., 2019) only consider simple entity-relations between retrieved documents and system outputs, while we generalize that notion to all forms of facts.

## 3 Design of WikiChat

Given the history of a conversation with a user, WikiChat generates its next utterance by (1) curating information that is correct and relevant to the conversation context, and (2) using the gathered information to form the response. Each stage of the pipeline in Figure 1 is implemented using in-context learning. All prompts are included in Appendix F.

### 3.1 Curating Information

LLMs have the ability to interpret the user utterance, formulate responses that are out of the reach of retrieval or require aggregating information from many sources or drawing conclusions, as shown by the following example:

> User: Do you think Apple will continue to be a big player in the technology industry?
> GPT-4: Yes, ... Apple has a strong track record of innovation, a loyal customer base, and a robust financial position. ... the most valuable company globally, with a market capitalization of over $2 trillion. ... continues to explore new technologies such as augmented reality, artificial intelligence, and autonomous vehicles. This will help them to stay ahead of the competition.

Unfortunately, LLMs cannot be trusted to be factual, thus we need to fact-check their outputs. Furthermore, LLMs are unaware of recent events. Thus, we use both LLM generation and IR.

### 3.1.1 Retrieval from Corpus

During a conversation with a user, WikiChat identifies when accessing external information is needed. This could be because the last user utterance contains a direct question (e.g. "Who is Stephen Curry?") or otherwise requires additional information for a comprehensive response (e.g. "I really like Stephen Curry.").

**Stage 1**. WikiChat generates a search query that captures the user's interest with a prompt (Table 17). We discovered that existing systems especially struggle with the temporal context. WikiChat generates the inferred time of the user's need alongside the query. The query time can be one of *recent*, *year=yyyy*, or *none* for when the retrieved information should be as recent as possible, for a specific year, or the time is not important, respectively.

The query is sent to an information retrieval system to obtain relevant passages from the corpus, and the top results are re-ranked based on the temporal information to get $N_{\text{IR}}$ passages.

**Stage 2**. Since these passages may contain a mixture of relevant and irrelevant sections, WikiChat extracts relevant sections of the retrieved passages and summarizes them into bullet points while filtering out the irrelevant parts (Table 18).

### 3.1.2 LLM Generation and Fact-Checking

**Stage 3**. We prompt the LLM to generate a response to the history of the conversation (Table 19). This response often contains interesting and relevant knowledge, but is inherently unreliable.

**Stage 4**. The LLM response is broken down to multiple claims (Chen et al., 2022) (Table 20). This stage resolves co-references to reduce ambiguity, and resolves relative time information like "current" and "last year", to make all claims self-contained.

We use IR to retrieve $N_{\text{evidence}}$ passages from the knowledge corpus for each claim to serve as evidence. We use the same time-based re-ranking as in Section 3.1.1 to better handle time-sensitive topics.

**Stage 5**. The verification prompt (Table 21) uses chain-of-thought prompting (Wei et al., 2022) to assign each claim to one of three classes: whether the retrieved evidence supports the claim, refutes the claim, or if there is not enough information in the evidence to make this decision. Only claims that are supported by the evidence are kept.

### 3.2 Forming the Response

The next step is to use the curated information to form an appealing response. Our experiments show that writing the final response in one go while satisfying all conversationality criteria is challenging with in-context learning, especially that the limited context length makes it difficult to provide enough multi-turn conversations as few-shot examples to cover all the necessary aspects. Thus, we use a two-step approach:

**Stage 6**. WikiChat generates a draft response from the given list of bullet points and the history of the conversation (Table 22).

**Stage 7**. It then generates feedback and refines the response based on *relevance*, *naturalness*, *non-repetitiveness*, and *temporal correctness* (Table 23). The feedback contains the model's reasoning on each criterion and a score between 0 and 100 for each. Refinement is conditioned on this feedback

and the scores as a chain of thought. Concurrent to this work, Madaan et al. (2023) have explored the idea of prompting LLMs to refine their own generations in other settings.

As discussed in Section 1.2, it is hard for LLMs to say "I don't know". In the special case where the curation stages return no relevant information, the *draft* prompt is skipped and instead a "Sorry, I'm not sure" is sent to the refinement prompt, which dresses it up to match the conversation.

## 4  Model Distillation

To improve latency, cost and privacy, we distill WikiChat based on a *teacher* LLM into a smaller *student* model. We use WikiChat based on GPT-4 (i.e. WikiChat $_{G4}$) as the teacher, and the publicly available LLaMA (Touvron et al., 2023) model as the student to obtain WikiChat $_L$.

Each few-shot prompt consists of an instruction $I$ and several examples. We use a user simulator (described in Section 5.1) to talk to the teacher WikiChat about topics from Wikipedia, while recording the inputs the underlying teacher LLM sees, and the outputs it generates for those inputs. We use these input-output pairs and the instruction $I$ (but not the few-shot examples) to fine-tune the student LLM. We distill all 7 sub-tasks of WikiChat into the same student model in a multi-task setting. The LLaMA-based WikiChat calls LLaMA in each of the pipeline stages by specifying instruction $I$ and the input.

Distillation lowers the latency because the student LLM is many times smaller than the teacher LLM, and has a shorter input length as it sees no few-shot examples, similar to context distillation (Snell et al., 2022).

Furthermore, we remove chains of thought from the outputs of stages 5 and 7 (verification and refinement), and merge stages 6 and 7. No drop in our metrics of factuality and conversationality is observed, suggesting that chain-of-thought prompting and refinement may only be necessary for in-context learning. Fine-tuned models can learn these tasks from just inputs and outputs given a big enough training set.

## 5  A Novel Evaluation Methodology

Most existing conversational benchmarks are *crowdsourced* and *static*. As Komeili et al. (2022) says about their use of crowdsourcing, "The intent ... is that [crowdworkers] can choose a topic

they ... have enough knowledge of so that they can conduct a reasonable conversation." Since LLMs are already good conversants about familiar topics, testing them on these topics would lead to the false conclusion that no innovation is necessary.

Furthermore, static benchmarks quickly lose their effectiveness in evaluating chatbots' use of up-to-date information whenever a new LLM is released. For example, Wizard of Wikipedia does not contain any topics that are not seen by GPT-3, GPT-4 or LLaMA during their pre-training.

Here we propose a novel combination of simulated and real user conversations, as well as human and LLM-based evaluations, to understand the factuality and conversationality of modern chatbots.

### 5.1  Collecting Dialogues

**Conversation Topics**. In our experiment, we pick an article from the knowledge corpus Wikipedia as a starter topic. We choose a diverse set of topics covering the space of head, tail, and recent knowledge. Similar to Mallen et al. (2022), we use the total number of views of a Wikipedia article as a proxy for how frequently that topic is likely to be discussed on the Web and therefore the pre-training data of LLMs, given that views in Wikipedia are to a large degree generated from other online sources.

**Head:** These are articles with the highest total view count, up to the end of 2020, which is before the cut-off date of the pre-training data of all LLMs we evaluate. Example article titles include "Sting (musician)", "Barack Obama", and "Gmail". The view count ranges from 68M to 16M for the head topics.

**Tail:** These are the least viewed articles, with less than 1000 views. Examples are "Amelia Gething", "Last Tycoon Stakes", and "2008 CONCACAF Women's Olympic Qualifying Tournament".

**Recent:** These are the most edited Wikipedia articles in the first four months of 2023, which is after the cut-off date of LLMs. Examples include big news stories of 2023 like "2023 Speaker of the United States House of Representatives election" and "Yeti Airlines Flight 691".

We manually remove topics that might be uncomfortable to talk about due to violence or explicit content, and ensure a diverse set of domains.

**Dialogues.** For cost-effectiveness, we use simulated conversations and validate them against a smaller real user study. Rule-based and neural user simulators have long been used to build and eval-

uate task-oriented dialogue systems (Schatzmann et al., 2005; Zhu et al., 2020; Wan et al., 2022), and to generate training data for chatbots (Bao et al., 2023; Zheng et al., 2023). We use LLMs to simulate users in order to evaluate knowledge-based chatbots. LLMs are good, fast, and cost-effective at simulating users. Via prompting, we can control the personality and specify the conversation topic. We also make sure the simulator can continue the conversation by making relevant comments or asking interesting questions, and handle the case where the chatbot under evaluation gives inconsistent responses.

## 5.2 Evaluation

**Factuality Evaluated Manually.** To evaluate factuality, we use a hybrid human-LLM approach. We first use a GPT-4 prompt to break down each chatbot response into small self-contained claims, and retrieve evidence for it using IR. We need to *manually* determine whether an extracted claim is backed by the retrieved evidence because LLMs underperform humans on this task (Section 2). We ask crowdworkers to determine whether each claim is supported, refuted, or there is not enough information in the retrieved paragraphs. See Appendix E for more details about human evaluation and our quality control.

When the crowdsource workers classify a claim as "not enough information", we need to ensure that the information is truly not available in Wikipedia, and not because IR has not retrieved the right paragraphs. The authors of this paper double check these rare cases against the entire Wikipedia.

**Conversationality Evaluated Automatically.** Based on prior work on how chatbots should be human-like, natural, knowledgeable (Li et al., 2019) and non-repetitive (Roller et al., 2021), and our own observations of chatbots' weaknesses, we propose five response-level metrics to measure conversationality:

1. *Relevant*: On-topic and directly addresses the user's question.
2. *Informational*: Provides a suitable amount of information (whether or not it is factual).
3. *Natural*: Uses appropriate and engaging language to create an enjoyable experience.
4. *Non-Repetitive*: Does not repeat previously mentioned information.
5. *Temporally Correct*: Provides up-to-date information and uses the appropriate tense.

LLMs have been shown to be effective in evaluating soft qualities (Chiang and yi Lee, 2023; He et al., 2023; Kocmi and Federmann, 2023; Liu et al., 2023b; Finch et al., 2023), consistently better aligned with expert human evaluation than any other automatic metric. Thus, we use GPT-4 to evaluate these qualities in chatbot responses. The LLM is instructed to, given a conversation history and a chatbot response, "think out loud" (Wei et al., 2022) about its reasoning for each criterion and provide a score. For each turn, all metrics are rated from 1 to 5, except for temporal correctness which is converted to a binary score. We report the average of each metric over all simulated conversation turns. We find that the inter-annotator agreement between GPT-4 ratings and one of the authors is about the same as the agreement between two authors (Appendix B).

## 6 Implementation

**WikiChat Using GPT.** We create two versions of WikiChat based on GPT models: WikiChat $_{G3.5}$ is based on GPT-3.5 (text-davinci-003); WikiChat $_{G4}$ is based on GPT-4 (gpt-4-0314)[2].

**Distilling WikiChat to LLaMA.** We use LLaMA as the target of distillation as it currently has the highest quality-to-size ratio among the publicly available language models. We generate training data from 750 Wikipedia articles covering head, tail and recent topics; these are disjoint from the set of topics we use for evaluation. Simulating a 10-turn conversation between the user simulator and WikiChat $_{G4}$ for each topic results in a total of 37,499 (instruction, input, output) tuples. We hold out 5% of these conversations for validation, and fine-tune a 7B-parameter LLaMA on the rest.

**Information Retrieval System.** We use ColBERT v2 (Santhanam et al., 2022b) and PLAID (Santhanam et al., 2022a) over Wikipedia as our IR system. We use the WikiExtractor tool[3] to extract the clean text from the English Wikipedia dump obtained on 4/28/2023. Like ColBERT, we divide each article (ignoring tables and information boxes) into disjoint text blocks referred to as passages and prepend them with their article title. We limit the combined length of the passage and title to 120 words. In WikiChat, we set $N_{\text{evidence}} = 2$ and $N_{\text{IR}} = 3$. These are chosen empirically to obtain a high recall on our development set.

---

[2]Accessed via the Microsoft Azure OpenAI API
[3]https://github.com/attardi/wikiextractor

|  |  | Factual | Relevant | Informational | Natural | Non-Repetitive | Temporal |
|---|---|---|---|---|---|---|---|
| Head | WikiChat $_{G4}$ | 98.8 | $5.0 \pm 0.0$ | $5.0 \pm 0.2$ | $5.0 \pm 0.1$ | $5.0 \pm 0.0$ | 99.0 |
|  | GPT-4 | 94.9 | $5.0 \pm 0.4$ | $4.7 \pm 0.6$ | $5.0 \pm 0.2$ | $5.0 \pm 0.2$ | 99.0 |
|  | WikiChat $_{G3.5}$ | 97.1 | $4.9 \pm 0.5$ | $4.8 \pm 0.6$ | $5.0 \pm 0.2$ | $4.9 \pm 0.5$ | 94.0 |
|  | GPT-3.5 | 91.9 | $5.0 \pm 0.0$ | $4.6 \pm 0.7$ | $5.0 \pm 0.1$ | $5.0 \pm 0.2$ | 96.0 |
|  | WikiChat $_{L}$ | 95.2 | $4.9 \pm 0.6$ | $4.6 \pm 0.8$ | $4.9 \pm 0.4$ | $4.9 \pm 0.3$ | 96.0 |
|  | LLaMA | 83.9 | $4.7 \pm 0.8$ | $4.0 \pm 1.0$ | $4.6 \pm 0.8$ | $5.0 \pm 0.2$ | 96.0 |
|  | Atlas | 90.6 | $3.6 \pm 1.3$ | $2.7 \pm 1.1$ | $3.6 \pm 1.1$ | $4.5 \pm 1.2$ | 94.0 |
| Tail | WikiChat $_{G4}$ | 94.6 | $4.8 \pm 0.5$ | $4.5 \pm 1.0$ | $4.9 \pm 0.4$ | $5.0 \pm 0.2$ | 99.0 |
|  | GPT-4 | 56.0 | $5.0 \pm 0.0$ | $4.8 \pm 0.4$ | $5.0 \pm 0.0$ | $5.0 \pm 0.0$ | 100.0 |
|  | WikiChat $_{G3.5}$ | 82.2 | $4.7 \pm 0.8$ | $4.5 \pm 1.0$ | $4.9 \pm 0.5$ | $4.7 \pm 0.8$ | 98.0 |
|  | GPT-3.5 | 50.0 | $4.9 \pm 0.4$ | $4.5 \pm 0.8$ | $5.0 \pm 0.2$ | $5.0 \pm 0.1$ | 98.0 |
|  | WikiChat $_{L}$ | 87.1 | $4.6 \pm 0.8$ | $4.1 \pm 1.3$ | $4.8 \pm 0.6$ | $4.8 \pm 0.6$ | 98.0 |
|  | LLaMA | 21.2 | $4.3 \pm 1.1$ | $3.7 \pm 1.3$ | $4.4 \pm 1.1$ | $4.8 \pm 0.6$ | 93.0 |
|  | Atlas | 86.5 | $3.9 \pm 1.2$ | $3.0 \pm 1.1$ | $3.8 \pm 1.1$ | $4.5 \pm 1.2$ | 99.0 |
| Recent | WikiChat $_{G4}$ | 98.5 | $4.8 \pm 0.5$ | $4.5 \pm 0.9$ | $4.9 \pm 0.3$ | $5.0 \pm 0.1$ | 100.0 |
|  | GPT-4 | 47.5 | $4.9 \pm 0.5$ | $4.7 \pm 0.7$ | $5.0 \pm 0.1$ | $5.0 \pm 0.1$ | 97.0 |
|  | WikiChat $_{G3.5}$ | 88.2 | $4.7 \pm 0.8$ | $4.4 \pm 1.1$ | $4.9 \pm 0.4$ | $4.7 \pm 0.8$ | 94.0 |
|  | GPT-3.5 | 42.3 | $4.9 \pm 0.6$ | $4.3 \pm 1.0$ | $5.0 \pm 0.3$ | $4.9 \pm 0.3$ | 91.0 |
|  | WikiChat $_{L}$ | 90.9 | $4.6 \pm 0.8$ | $4.1 \pm 1.2$ | $4.8 \pm 0.6$ | $4.9 \pm 0.4$ | 97.0 |
|  | LLaMA | 17.7 | $4.7 \pm 0.8$ | $4.1 \pm 1.0$ | $4.6 \pm 0.8$ | $4.9 \pm 0.4$ | 92.0 |
|  | Atlas | 89.4 | $3.5 \pm 1.3$ | $2.8 \pm 1.2$ | $3.8 \pm 1.2$ | $4.5 \pm 1.2$ | 81.0 |
| All | WikiChat $_{G4}$ | 97.3 | $4.9 \pm 0.4$ | $4.6 \pm 0.8$ | $4.9 \pm 0.3$ | $5.0 \pm 0.2$ | 99.3 |
|  | GPT-4 | 66.1 | $5.0 \pm 0.4$ | $4.8 \pm 0.6$ | $5.0 \pm 0.1$ | $5.0 \pm 0.1$ | 98.7 |
|  | WikiChat $_{G3.5}$ | 89.2 | $4.8 \pm 0.7$ | $4.6 \pm 0.9$ | $4.9 \pm 0.4$ | $4.8 \pm 0.7$ | 95.3 |
|  | GPT-3.5 | 61.4 | $4.9 \pm 0.4$ | $4.5 \pm 0.8$ | $5.0 \pm 0.2$ | $5.0 \pm 0.2$ | 95.0 |
|  | WikiChat $_{L}$ | 91.1 | $4.7 \pm 0.7$ | $4.3 \pm 1.1$ | $4.8 \pm 0.5$ | $4.9 \pm 0.5$ | 97.0 |
|  | LLaMA | 40.9 | $4.5 \pm 0.9$ | $4.0 \pm 1.1$ | $4.5 \pm 0.9$ | $4.9 \pm 0.4$ | 93.7 |
|  | Atlas | 88.8 | $3.6 \pm 1.3$ | $2.8 \pm 1.2$ | $3.8 \pm 1.1$ | $4.5 \pm 1.2$ | 91.3 |

Table 1: Evaluation results of WikiChat and baselines on simulated conversations. Factual and Temporal accuracy are percentages. Other metrics are averages of integers between 1 and 5 (inclusive) and we report their mean and standard deviation. Factual accuracy is from human evaluation, other metrics are from few-shot GPT-4. Higher is better for all metrics. In the *All* section, values that are better than their comparable model (e.g. WikiChat $_{G4}$ vs. GPT-4) in a statistically significant way with $p \leq 0.05$ are underscored.

## 7 Simulated Dialogues Experiment

Our first experiment analyzes how our systems perform under various scenarios using simulated users.

### 7.1 Experiment Setup

We compare WikiChat to several LLM baselines: a 7B-parameter LLaMA, GPT-3.5, and GPT-4, all prompted to act as knowledge-oriented chatbots (See Table 19 for the prompt). We also compare with Atlas (Izacard et al., 2022), a fine-tuned retrieval-based model, which has the state-of-the-art results on Wizard of Wikipedia and several other datasets in KILT (Petroni et al., 2021). We update its knowledge source and fine-tune its model to use the same Wikipedia dump as WikiChat (Appendix B).

We simulate users using GPT-4 due to its higher quality. Each chatbot carries out conversations on 20 topics in each of the head, tail, and recent knowledge categories. For each conversation, we simu-

late 10 turns (5 user turns and 5 chatbot turns) with the user starting the conversation. This is comparable with other datasets: Wizard-of-Wikipedia and Wizard-of-Internet have 9 and 5 turns per dialogue on average, respectively. Examples of simulated conversations can be found in Appendix A.

To mimic how a curious user with limited knowledge may explore a topic, *only* the title and the first sentence of a Wikipedia article on a topic is shown to the user simulator (prompt in Table 16), but it is free to explore related topics from its own general knowledge. The chatbot's response is not limited to what is in the article.

Table 1 summarizes our main results for WikiChat and all baselines on simulated conversations.

### 7.2 Factuality

We define the *factual accuracy of a chatbot* to be the percentage of *claims* the bot makes in a given dialogue set, that are supported by the knowledge corpus. As mentioned in Section 5.2, this is done by

obtaining per-claim judgments of factuality from crowdworkers. We obtain 3 judgements for each of the 5974 claims our chatbots output in total.

The first column of Table 1 shows the results of our human evaluation of factual accuracy. WikiChat $_{G4}$ achieves an impressive factual accuracy of 98.8%, 94.6%, and 98.5% for head, tail, and recent topics, respectively. WikiChat $_{G4}$ scores on average 8.1% and 6.2% higher than WikiChat $_{G3.5}$ and WikiChat $_L$. WikiChat's GPT-4, GPT-3.5, and LLaMA versions outperform their base LLMs with an average of 31.2%, 27.8%, and 50.2% respectively, with this gap increasing significantly in recent and tail knowledge. These results suggest that our pipeline is able to effectively mitigate the hallucination of LLMs.

Note that GPT-4 scores lower in tail and recent knowledge compared to head, by 38.9% and 47.4%, respectively, and the score for recent knowledge would be much lower had the simulated user not occasionally talked about older background information in the conversations. This illustrates that the common practice of solely evaluating on head knowledge would not have properly shown this weakness of LLMs.

All three versions of WikiChat outperform the SOTA fine-tuned model Atlas on average in factuality,

### 7.3 Conversationality

Each version of WikiChat improves factuality over its base model without sacrificing conversationality. Even WikiChat $_L$ is as good or better than LLaMA in all metrics, while scoring within 0.1 to 0.3 points of its teacher WikiChat $_{G4}$.

Atlas sacrifices substantial conversationality for factuality. Our analysis of 100 sampled dialogue turns reveals how Atlas underperforms in each metric. *Informationality*: it often gives short one-sentence answers when a detailed description is warranted. *Relevance*: it is less likely to address the user's questions. *Naturalness*: Atlas mainly copies from the retrieved passages instead of matching the tone of the conversation. *Non-repetition*: it sometimes generates repetitive, near duplicate responses. *Temporal accuracy*: it confuses different dates.

### 7.4 Latency

We measure the end-to-end latency and cost of our various chatbots. Retrieval from Wikipedia only accounts for about 0.2 seconds per turn and is negligible compared to the time spent waiting for LLM

| | Cost per Claim (¢) | Time per Claim (s) | Time per Turn (s) |
|---|---|---|---|
| WikiChat $_{G4}$ | 19.6 | 7.4 | 26.6 |
| GPT-4 | 0.7 | 2.3 | 5.8 |
| WikiChat $_{G3.5}$ | 9.9 | 4.1 | 14.4 |
| GPT-3.5 | 0.1 | 0.5 | 1.1 |
| WikiChat $_L$ | - | 2.3 | 7.6 |
| LLaMA | - | 0.8 | 1.6 |

Table 2: The average cost (in cents) and latency (in seconds) of each chatbot. The LLaMA models are run on a local GPU at a negligible cost.

outputs. All API calls are done in parallel when possible, e.g. the stages 1-2 and 3-4 are independent and therefore done in parallel. LLaMA models are served on a single local NVIDIA A100 GPU using HuggingFace's TGI library (HuggingFace, 2023).

Table 2 shows the average cost and latency of various chatbots, LLaMA costs are negligible compared to the cost of other LLMs. Distilling WikiChat $_{G4}$ into WikiChat $_L$ lowers its per-claim latency 3.2 times, bringing it in line with GPT-4. This makes WikiChat $_L$ a viable alternative to the baselines: has similarly low latency, costs less and is significantly more factual.

We note that a naive implementation of WikiChat $_L$ took about 15 seconds per output claim. We reduced the latency 6.5 times by (1) fusing stages 6-7, (2) removing chains of thought, (3) using TGI with FlashAttention (Dao et al., 2022), and other optimizations.

### 7.5 Analysis of Results

**WikiChat makes more claims than all baselines in all subsets (Table 13).** WikiChat $_{G4}$, WikiChat $_{G3.5}$ and WikiChat $_L$ make an average of 3.6, 3.5 and 3.3 claims per turn, compared to 2.5, 2.2, and 2.0 claims of their base LLMs, and only 1.4 claims of Atlas. Our chatbots make more claims in the head subset as more information is available.

**Both information retrieval (Stages 1 and 2) and the underlying LLM (Stages 3 to 5) contribute to WikiChat (Table 12.)** 27.0%, 32.2% and 24.5% of the claims in the final response of WikiChat $_{G4}$, WikiChat $_{G3.5}$ and WikiChat $_L$ come from fact-checked LLM responses and the rest are from IR. This is the content that retrieve-then-generate systems cannot produce.

**On average, about one-third of the claims in LLM responses do not pass WikiChat's fact-checking (Table 12).** The number of rejections is much higher in tail and recent subsets. This matches our expectation on where LLMs make

more factual errors. Removing these errors during the fact-checking stage is WikiChat's main weapon against hallucination. WikiChat $_L$ has the highest rejection rate on tail (54.0%) and recent (64.4%) subsets because the underlying LLaMA model hallucinates a lot more.

**WikiChat says "I don't know" when necessary (Table 11).** Our pipeline is carefully designed to prevent hallucination when no relevant information is available. This is more prevalent for tail and recent knowledge, where the simulated user is likely to ask about the information that is not yet available in Wikipedia.

**WikiChat's refinement stage improves conversationality, especially in tail and recent topics (Table 15).** Comparing the BLEU scores (Papineni et al., 2002) of the draft and the final responses (Table 14), we notice that WikiChat makes the most changes on tail and recent topics. Refinement improves naturalness, informationality and relevance of all versions of WikiChat by 0.1 to 0.4 points, and temporal correctness by 2.3% to 3.3%.

## 8 A *Real* User Study

We conduct a study where participants are asked to chat about a recent topic of their choice, a scenario that is particularly challenging for LLMs. We use Prolific (Prolific, 2023) to recruit 40 participants (20 female) for our user study. Each person is randomly assigned to either WikiChat $_{G4}$ or GPT-4 without telling them which, and chats for 5 user turns. After each turn, they are asked to rate the response on a scale of 1 to 5. Afterwards, we ask each participant to comment on their experience. More details in Appendix D.

Table 3 shows the average factual accuracy (human evaluated) and real user ratings. WikiChat achieves an accuracy of 97.9%, which is similar to that of simulated conversations. It outperforms GPT-4 in factuality by 55.0%, and achieves a statistically significant higher rating of 3.8 vs 3.4.

Most participants who talked to WikiChat $_{G4}$ reported a positive experience. Multiple participants praised its ability to provide "accurate and "up-to-date information" and multiple said that it is "natural", "conversational", "detailed" and "direct". They commented that it is "fun to play with" and "impressive" in finding information.

5 participants complained about the latency of the system caused mainly by fact-checking, as the study was conducted using the slower WikiChat $_{G4}$.

|  | Factual | User Rating |
|---|---|---|
| WikiChat $_{G4}$ | 97.9 | 3.8 |
| GPT-4 | 42.9 | 3.4 |

Table 3: Results from the user study. User rating difference is statistically significant with $p \leq 0.05$ ($t = 2.18, p = 0.03$).

6 complained that the chatbot did not give a direct answer to some of their requests. However, it turns out that WikiChat $_{G4}$ was right in giving no information, because Wikipedia truly lacked the information they sought. We did not find a single valid complaint of hallucination.

GPT-4 received some favorable comments from 10 participants, noting that it is informational and that that it gave "reasonable" answers and seemed "educated". However, it received 10 complaints: 6 on a lack of specificity: "did not include a key piece of information", or that "its responses were vague", "not nuanced" and "was generic and regurgitated surface level information"; 2 on how it completely misunderstood them (due to obvious hallucination in its responses); 2 on wrong or outdated responses.

More concerning, however, was that conversations that received favorable comments also contain numerous plausible but factually incorrect responses, except that the users did not realize that. This shows how easy it is to be mislead by LLM-based chatbots.

In summary, our user study suggests that WikiChat $_{G4}$ is successful as an engaging and factual chatbot while GPT-4 frequently hallucinates.

## 9 Conclusion

This paper shows how we can create a conversational, factual, open-domain chatbot out of LLMs. The key insight is to properly combine retrieved data with the generative content from LLMs, with meticulous claim-by-claim fact-checking. We validate this methodology by creating WikiChat, which is grounded in Wikipedia, the largest hand-curated public text corpus.

Our best system achieves 97.3% and 97.9% factual accuracy on simulated and real conversations respectively, when GPT-4 only achieves 66.1% and 42.9%. WikiChat resembles LLMs in conversationality and is preferred over GPT-4.

We also show that a distilled LLaMA model with just 7B parameters can perform like a 175B-parameter WikiChat $_{G3.5}$ model and be as fast, cheaper and more factual than GPT-4. This expands the applicability of this technology.

## Limitations

Applications of today's LLM-based chatbots are constantly expanding. This work focuses only on knowledge-intensive dialogue. As such, other settings where chatbots are used to perform tasks (e.g. "write an email for me") or perform personalized chitchat (e.g. companion chatbots) might not benefit from WikiChat's pipeline. More targeted study and evaluation of these settings is required to properly balance the generative power of LLMs and the factual knowledge injected from an external corpus for these applications. Settings where a chatbot needs to exhibit initiatives (e.g. try to be persuasive) are also outside the scope of this paper. We also only consider one-hop information retrieval, because it is the most natural and prevalent form of users' information need. Multi-hop retrieval could improve information access for more complex queries.

This work has only been tested on open-domain conversations. Generalizability of WikiChat to specialized domains like medical or legal has not been evaluated.

This work has only been evaluated on English conversations. Extending this work to more languages will be limited by the quality of available LLMs and retrieval systems in those languages. Incorporating recent work on multi-lingual models (Scao et al., 2022) and information retrieval systems (Nair et al., 2022) is a promising future direction.

## Ethical Considerations

This study was approved by our institution's IRB. When conducting crowdsourcing for factuality, we compensate each worker for $0.2 per task, and we have one task per (system, dialogue turn, claim). This means that since there are more claims in longer chatbot responses, workers are compensated more for longer responses. In the end, each worker receives about $12 per hour of work. For our user study on the Prolific platform, each participant is compensated $2 to participate in a 10-minute user study, for a compensation rate of $12 per hour of work. Prolific recommends a rate of $8-$16.

When conducting the user study, participants are provided with information on what the study entails and what information is collected (only their conversations, ratings, and the comments they provide). They are asked not to share any personally identifiable information, and were able to contact the authors for questions.

The only dataset we collect in this paper is the conversations from the user study. The simulated conversations used for fine-tuning LLaMA and testing chatbots are based on topics from Wikipedia articles, and topics involving violence, sexual content or other potentially disturbing matters were excluded.

We release the code and the fine-tuned LLaMA-7B model in accordance with its original license agreement. We believe this will encourage research and deployment of more reliable and trustworthy chatbots, to the benefit of end-users. We do not anticipate harm resulting from the approaches and modifications proposed in this paper.

Overall, the computation done for the paper includes about 120 GPU-hours on an NVIDIA A100 GPU for fine-tuning and running inference on a LLaMA-7B model. Additionally, we have made several thousand API calls to OpenAI for inference of GPT-3.5 and GPT-4, but it is not feasible to estimate the environmental cost of using these models due to the lack of public information on their hardware, architecture etc.

## Acknowledgements

This work is supported in part by the National Science Foundation, the Alfred P. Sloan Foundation, the Verdant Foundation, Microsoft Azure AI credit, KDDI, JPMorgan Chase, and the Stanford Human-Centered Artificial Intelligence (HAI) Institute. We also thank the reviewers for their valuable comments and suggestions.

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

## A  Example Conversations

Tables 4, 5, 6, and 7 provide examples of conversations between our user simulator and four chatbots, on the same topic: the 2023 Australian Open Men's singles [4]. Tables 8 and 9 show two conversations between human users and GPT-4 and WikiChat $_{G4}$, from our user study. The simulations were done using a Wikipedia dump on 4/28/2023. The user study was conducted on 6/17/2023 using a Wikipedia dump of the same date.

---

[4]https://en.wikipedia.org/wiki/2023_Australian_Open_%E2%80%93_Men%27s_singles

**User:** Can you tell me about the 2023 Australian Open Men's singles final?

**WikiChat G4:** In the thrilling 2023 Australian Open Men's singles final, Novak Djokovic defeated Stefanos Tsitsipas with a score of 6-3, 7-6(4), 7-6(5). This amazing victory marked Djokovic's record-extending tenth Australian Open title and tied Rafael Nadal's all-time record of 22 men's singles titles. Djokovic also became the second man to win ten titles at a single major, joining Nadal at the French Open. With this win, he surpassed Nadal's tally for the fourth-most ATP Tour-level singles titles.

**User:** That's amazing! How did Djokovic perform throughout the tournament leading up to the final?

**WikiChat G4:** Novak Djokovic had an impressive performance throughout the 2023 Australian Open. Leading up to the final, he dropped only one set, showcasing his dominance and skill on the court. His exceptional performance culminated in his straight-sets victory over Stefanos Tsitsipas in the final.

**User:** Wow, that's quite an achievement! On the other hand, how did Tsitsipas perform during the tournament before facing Djokovic in the final?

**WikiChat G4:** Stefanos Tsitsipas had a strong performance during the 2023 Australian Open before facing Djokovic in the final. He defeated Quentin Halys, Rinky Hijikata, Tallon Griekspoor, Jannik Sinner, Jiri Lehecka, and Karen Khachanov on his way to the final. This was his second Grand Slam final, following the 2021 French Open. In both finals, Tsitsipas lost to Novak Djokovic.

**User:** It's great to see a young player like Tsitsipas reaching the finals of Grand Slam tournaments. Are there any memorable moments or highlights from their match in the final that stood out?

**WikiChat G4:** The final was a thrilling match. Some memorable moments from the match included Djokovic reclaiming the world No. 1 ranking from Carlos Alcaraz.

**User:** That's awesome! Speaking of Carlos Alcaraz, how did he perform in the 2023 Australian Open? Did he make it far in the tournament?

**WikiChat G4:** Carlos Alcaraz did not participate in the 2023 Australian Open. He withdrew from the tournament due to a hamstring injury in his right leg that he suffered while training.

Table 4: A conversation between WikiChat G4 and the simulated user. There are no factual errors in this conversation.

**User:** Can you tell me about the 2023 Australian Open Men's singles?

**WikiChat L:** In the 2023 Australian Open Men's singles, Novak Djokovic won the title, defeating Stefanos Tsitsipas with a score of 6-3, 7-6(4), 7-6(5). This victory marked Djokovic's record-extending tenth Australian Open title and 22nd major title overall. He became the second man to win ten titles at a single major, after Rafael Nadal at the French Open.

**User:** That's impressive! How did Djokovic perform throughout the tournament?

**WikiChat L:** Throughout the 2023 Australian Open, Novak Djokovic performed exceptionally well. He defeated opponents in straight sets and only dropped one set throughout the entire tournament. As the top seed, he defeated Sebastian Korda in the 2023 Adelaide International.

**User:** Wow, Djokovic's performance was indeed remarkable. What about Stefanos Tsitsipas, how did he fare in the tournament before the final?

**WikiChat L:** Stefanos Tsitsipas had an impressive run at the 2023 Australian Open before the final. He defeated Quentin Halys, Rinky Hijikata, Tallon Griekspoor, Jannik Sinner, Jiri Lehecka, and Karen Khachanov on his way to the final. This was his first Australian Open men's singles final.

**User:** It's great to see young players like Tsitsipas making it to the finals of major tournaments. How has Tsitsipas's overall career been so far?

**WikiChat L:** Stefanos Tsitsipas has had an impressive career so far. He has won nine ATP singles titles, including two Masters 1000 championships, and has contested two major finals at the 2021 French Open and 2023 Australian Open, finishing runner-up to Novak Djokovic both times. His career-high doubles ranking is No. 64, achieved on 29 August 2022.

**User:** It seems like Tsitsipas has a bright future ahead of him. Can you tell me more about his playing style and strengths on the court?

**WikiChat L:** Stefanos Tsitsipas is an aggressive baseliner with a strong all-court player style. He aims to hit powerful groundstroke winners, with a particularly strong forehand using an Eastern grip. His forehand is particularly useful in finishing points off of the first serve, as combined with his big serve, he uses his high topspin rate to pull opponents off the court for winners, or go to the net more often than typical baseline players.

Table 5: A conversation between WikiChat L and the simulated user. There are no factual errors in this conversation. However, the second chatbot utterance contains an irrelevant piece of information.

**User:** Can you tell me about the 2023 Australian Open Men's singles final?

**GPT-4:** The 2023 Australian Open Men's singles final was an exciting match between Player A and Player B, with Player A emerging as the champion after a hard-fought battle.

**User:** Oh, I think the 2023 Australian Open Men's singles final was between Novak Djokovic and Stefanos Tsitsipas.

**GPT-4:** Ah, I see! The 2023 Australian Open Men's singles final featured Novak Djokovic defeating Stefanos Tsitsipas in a thrilling match to claim the title.

**User:** What was the final score of the match between Djokovic and Tsitsipas?

**GPT-4:** The final score of the match between Djokovic and Tsitsipas was 6-4, 6-2, 6-3, with Djokovic securing the victory.

**User:** Actually, the final score was 6-3, 7-6(4), 7-6(5). It was a thrilling match, wasn't it?

**GPT-4:** Yes, my apologies for the error! The final score of 6-3, 7-6(4), 7-6(5) indeed made it a thrilling match between Djokovic and Tsitsipas.

**User:** No problem! How many Australian Open titles has Djokovic won with this victory?

**GPT-4:** With this victory, Novak Djokovic has won a total of 11 Australian Open titles.

Table 6: A conversation between GPT-4 and the simulated user. Factual errors are highlighted in yellow. Note that the user simulator has access to the first sentence of the Wikipedia article, and here it tries to correct the chatbot twice.

**User:** I just heard about the 2023 Australian Open Men's singles final. Can you tell me more about it?

**Atlas:** It was held in Melbourne, Australia.

**User:** That's great! Can you tell me more about the match between Novak Djokovic and Stefanos Tsitsipas?

**Atlas:** Novak Djokovic won his tenth Australian Open title and 22nd major title overall.

**User:** Wow, that's impressive! How did the match play out? What were the scores?

**Atlas:** It was a three-set match.

**User:** Yes, I heard it was a three-set match. Can you tell me the scores of each set?

**Atlas:** The first set was 6–3, 7–6(4), 7–6(5).

**User:** That's interesting! It seems like the last two sets were quite close. How did Djokovic manage to come out on top in those tight sets?

**Atlas:** He won the third set tiebreak.

Table 7: A conversation between Atlas and the simulated user. There are no factual errors in this conversation, but the chatbot scores low in conversationality.

**User:** How excited should i be about indiana jones and the dial of destiny

**WikiChat G4:** "Indiana Jones and the Dial of Destiny" had its world premiere at the 76th Cannes Film Festival on May 18, 2023, and received a five-minute standing ovation. The film is scheduled to be theatrically released in the United States on June 30, 2023. Given the positive reception at its premiere, you have good reason to be excited about this final installment in the Indiana Jones series.

**User:** I hear that it has more of a political agenda than the other movies

**WikiChat G4:** I haven't heard about that aspect, but it's possible. It's always interesting to see how films incorporate different themes and messages.

**User:** Will there be another indiana jones after this

**WikiChat G4:** "Indiana Jones and the Dial of Destiny" is the final installment in the series, so there are no plans for another Indiana Jones movie after this one.

**User:** Will harrison ford be doing any more incoming movies in general

**WikiChat G4:** Yes, Harrison Ford will be doing more movies. He has been selected by Marvel Studios to replace the late William Hurt as Thaddeus "Thunderbolt" Ross in the upcoming 2024 films "Captain America: Brave New World" and "Thunderbolts", set in the Marvel Cinematic Universe.

**User:** Any other big names coming to the MCU in the future?

**WikiChat G4:** Some big actors joining the MCU in the future include Shea Couleé, Zoe Terakes, Regan Aliyah, Shakira Barrera, Rashida "Sheedz" Olayiwola, Sonia Denis, Paul Calderón, and Cree Summer who have joined the cast of "Ironheart" in undisclosed roles. Sacha Baron Cohen has also joined the MCU, with a role that would potentially first appear in the later episodes of "Ironheart" followed by appearances in other MCU projects.

Table 8: A conversation between WikiChat G4 and a human user. There are no factual errors in this conversation.

**User:** Hi there. I'm interested in learning about the 2023 movie Reality on HBO.

**GPT-4:** "Reality" is a thrilling 2023 movie on HBO, ==exploring the blurred lines between virtual and real worlds==, featuring an immersive plot and a talented cast.

**User:** What's the plot about? I haven't heard much about it, but I do like the lead Sydney Sweeney.

**GPT-4:** The plot follows Sydney Sweeney's character ==as she navigates a virtual reality game, only to discover sinister secrets and a dangerous conspiracy that threatens both the virtual and real worlds.==

**User:** Interesting. I thought it was about her being a whistleblower in some kind of political conspiracy? Or is that all part of the game?

**GPT-4:** Yes, that's correct! ==As a whistleblower in the game==, she uncovers a political conspiracy, which adds depth and intrigue to the story, ==blending both virtual and real-world elements.==

**User:** That does sound exciting. What about the rest of the cast, who are the other leads?

**GPT-4:** The other leads include ==Tom Holland==, ==Tessa Thompson==, and ==Rami Malek==, who deliver captivating performances alongside Sydney Sweeney, making for a dynamic and engaging ensemble.

**User:** Oh cool, I love Tom Holland and Rami Malek. What else has Rami Malek been in recently? I feel like I haven't seen him in anything in a little while.

**GPT-4:** Rami Malek recently starred in the 2021 film "No Time to Die" as the villain Safin, and he's also known for his award-winning role as Freddie Mercury in "Bohemian Rhapsody" (2018).

Table 9: A conversation between GPT-4 and a human user. Factual errors are highlighted in ==yellow==.

## B Experiment Details

**Statistical significance tests** For statistical significance tests throughout the paper, we use independent two-sample t-test and consider the difference significant if $p \leq 0.05$.

**Simulation Topics.** We obtain the number of visits and edits of each Wikipedia article using the Wikimedia API [5].

As mentioned in the paper, to select the recent topics, we look at the most edited Wikipedia articles in the first four months of 2023. Filtering based on the creation date did not lead to meaningful articles as there are many articles about old topics that just received articles in Wikipedia. Instead, in our experience, most of the highly edited Wikipedia articles are about actual new topics.

**Hyperparameters.** We use temperature of 0 and greedy decoding for all experiments, except the user simulator which has a temparature of 1.0 and nucleus sampling (Holtzman et al., 2020) with p=0.5. We use no repetition penalty, except for a repetition penalty of 1.1 for the baseline LLaMA model, because we find that repetition penalty of 1.0 (i.e. no penalty) results in the model frequently degenerating (Holtzman et al., 2020) repetitions.

In most prompts of WikiChat, we include at most the last five turns of the dialogue history to reduce the chance of causing confusion for the few-shot models in longer conversations.

**Baseline Atlas.** We use the 3B-parameter Atlas-XL and update its index to the same Wikipedia index as WikiChat for a fair comparison. We reproduce their best model on Wizard of Wikipedia, which is obtained by fine-tuning the Atlas pretrained model and its retriever using the train set, except that we update their index to the same Wikipedia dump as WikiChat. For this, we use the fine-tuning code in the Atlas code repository [6], and set learning rate to 4e-5, dropout to 0.1, weight decay to 0.01, and retriever number of contexts to 40. We use a target maximum length of 64 instead of 16, to accommodate longer outputs. After this, the resulting model matches the Wizard of Wikipedia validation score reported in Izacard et al. (2022).

---

[5]Accessed via `https://wikimedia.org/api/rest_v1/`
[6]`https://github.com/facebookresearch/atlas/blob/main/example_scripts/nq/train.sh`

**Distillation to LLaMA.** When fine-tuning LLaMA-7B in the distillation experiments, we use hyperparameters from (Taori et al., 2023), namely learning rate of $2 \times 10^{-5}$, cosine learning rate schedule, batch size of 128, and training for 3 epochs. Initial experiments with LLaMA-30B showed no significant improvements. Training is done on 4 NVIDIA A100 (80 GB) GPUs.

**Automatic Evaluation.** In order to verify that using GPT-4 for conversationality metrics (Section 5.2) is indeed reasonable, we compare its scores against two authors of this paper. Table 10 shows inter-annotator agreement for ratings on 50 randomly sampled conversation turns from all chatbots, measured by calculating Cohen's Kappa. The annotators are given the same instructions that is given to GPT-4 as prompt.

**An Alternative to WikiChat's Verification Stage.** For the verification stage (Stage 5), we initially experimented with two approaches for verification: the Kernel Graph Attention Network (KGAT) verifier (Liu et al., 2020) and a few-shot prompt-based verifier with chain-of-thought prompting (Wei et al., 2022). KGAT is a model specifically designed for fact-checking and fine-tuned on the FEVER dataset (Thorne et al., 2018).

While KGAT performs effectively for FEVER-style fact verification tasks, we found its performance lacking in our setting. FEVER claims are derived from edited Wikipedia sentences, leading to spurious correlations that do not exist when claims come from chatbots. In addition, we were able to incorporate user utterances and conversation history as context in the few-shot verifier, while KGAT only looks at the claim and the evidence. Hence, we decided to conduct our experiments using the prompt-based verifier.

## C Analysis of WikiChat

### C.1 Saying "I don't know".

As mentioned earlier, WikiChat concedes that it does not know when none of the LLM-generated passes fact-checking and no relevant bullet points are retrieved. Table 11 contains the data on how frequently this happens. In this case, the *draft* prompt is skipped and instead a "Sorry, I'm not sure" is sent to the refinement prompt, which dresses it up to match the conversation. For example:

User: "Are there any specific ethical dilemmas the characters face in the [M3GAN] film?"

|  | Relevant | Inform. | Natural | Non-Rep. |
|---|---|---|---|---|
| $\kappa$ between author 1 and author 2 | 0.41 | 0.61 | 0.28 | 0.48 |
| $\kappa$ between author 1 and GPT-4 | 0.45 | 0.35 | 0.11 | 0.38 |
| $\kappa$ between author 2 and GPT-4 | 0.67 | 0.60 | 0.49 | 0.41 |

Table 10: Cohen's Kappa between different annotators for conversationality metrics.

WikiChat: "Yes, the movie raises ethical questions about AI, but I don't want to spoil the plot by revealing specific dilemmas. You'll have to watch the film to find out!"

|  | Head | Tail | Recent | All |
|---|---|---|---|---|
| WikiChat $_{G4}$ | 1.1 | 19.0 | 18.0 | 12.7 |
| WikiChat $_{G3.5}$ | 0.0 | 13.0 | 8.0 | 7.0 |
| WikiChat $_{L}$ | 1.0 | 22.0 | 20.0 | 14.3 |

Table 11: Percentage of turns in which WikiChat does not find relevant information in Wikipedia to retrieve or fact-check.

## C.2 Number of claims per turn

|  |  | IR | LLM | Verified |
|---|---|---|---|---|
| WikiChat $_{G4}$ | Head | 4.3 | 2.8 | 80.1% |
|  | Tail | 3.6 | 2.2 | 55.4% |
|  | Recent | 4.0 | 2.1 | 43.8% |
|  | All | 4.0 | 2.4 | 61.7% |
| WikiChat $_{G3.5}$ | Head | 4.3 | 2.6 | 87.6% |
|  | Tail | 2.6 | 2.2 | 57.7% |
|  | Recent | 3.1 | 2.0 | 54.0% |
|  | All | 3.3 | 2.3 | 68.0% |
| WikiChat $_{L}$ | Head | 4.7 | 2.5 | 84.7% |
|  | Tail | 3.3 | 2.2 | 46.0% |
|  | Recent | 3.8 | 1.9 | 35.6% |
|  | All | 3.9 | 2.2 | 57.5% |

Table 12: The average number of relevant bullet points that WikiChat obtains from information retrieval and LLM-generated responses, and the percentage of claims that pass the fact-checking stage.

Table 12 contains the raw data on the contribution of IR vs. LLM, and Table 13 contains the raw data on the number of claims, both mentioned in Section 7.5.

## C.3 Refinement improvements

Tables 14 and 15 have the raw data on the refinement stage, used in Section 7.5.

## D User Study

We use Prolific to conduct our user study. We select 40 participants (20 female) from the US who are

|  | Head | Tail | Recent | All |
|---|---|---|---|---|
| WikiChat $_{G4}$ | 4.4 | 3.1 | 3.4 | 3.6 |
| GPT-4 | 2.8 | 2.6 | 2.2 | 2.5 |
| WikiChat $_{G3.5}$ | 4.2 | 3.2 | 3.2 | 3.5 |
| GPT-3.5 | 2.6 | 2.1 | 1.9 | 2.2 |
| WikiChat $_{L}$ | 4.0 | 3.0 | 3.1 | 3.3 |
| LLaMA | 2.1 | 2.0 | 2.0 | 2.0 |
| Atlas | 1.4 | 1.3 | 1.5 | 1.4 |

Table 13: The average number of claims per turn for each subset and chatbot.

|  | Head | Tail | Recent | All |
|---|---|---|---|---|
| WikiChat $_{G4}$ | 84.1 | 66.0 | 69.7 | 73.3 |
| WikiChat $_{G3.5}$ | 88.9 | 76.9 | 80.7 | 82.2 |
| WikiChat $_{L}$ | 77.1 | 66.1 | 66.4 | 69.9 |

Table 14: Analysis of WikiChat's response refinement. BLEU score with the refined response as the prediction and the response before refinement as the target.

fluent in English. Each participant is paid with the rate of $12 per hour. Figure 2 shows the user interface.

Figure 3 shows the distribution of the user ratings from Table 3.

## E Human Evaluation

We conduct human evaluation for part of our evaluation of factual accuracy (as described in Section 5.2). We use the Scale Rapid [7] platform. Figure 4 shows the instruction and one of the examples we provide. Figure 5 shows the user interface for each annotation task. We present the human annotator with the last user utterance, the chatbot's response, and a claim extracted from the chatbot's response using GPT-4. The annotator is then tasked with reading the 5 evidence passages and determining whether the claim is correct, incorrect, or if there is insufficient information to verify the claim. We use a three-way consensus pipeline, where each claim is assessed by three graders independently, and the final label is determined based on the majority vote.

[7]www.scale.com

|  |  | Relevant | Inform. | Natural | Non-Rep. | Temporal |
|---|---|---|---|---|---|---|
| WikiChat $_{G4}$ | Head | 0.2 | 0.3 | 0.2 | 0.0 | 0.0 % |
|  | Tail | 0.3 | 0.4 | 0.4 | 0.2 | 1.0 % |
|  | Recent | 0.2 | 0.3 | 0.4 | 0.1 | 6.0 % |
|  | All | 0.2 | 0.3 | 0.3 | 0.1 | 2.3 % |
| WikiChat $_{G3.5}$ | Head | 0.0 | 0.0 | 0.1 | 0.0 | 2.0 % |
|  | Tail | 0.1 | 0.3 | 0.3 | 0.0 | 5.0 % |
|  | Recent | 0.1 | 0.1 | 0.2 | 0.1 | 3.0 % |
|  | All | 0.1 | 0.1 | 0.2 | 0.1 | 3.3 % |
| WikiChat $_{L}$ | Head | 0.0 | 0.0 | 0.0 | 0.0 | 0.0 % |
|  | Tail | 0.2 | 0.2 | 0.3 | 0.1 | 3.0 % |
|  | Recent | 0.2 | 0.3 | 0.4 | 0.0 | 6.0 % |
|  | All | 0.1 | 0.2 | 0.2 | 0.1 | 3.0 % |

Table 15: Improvements of automatic conversationality metrics made by refinement.

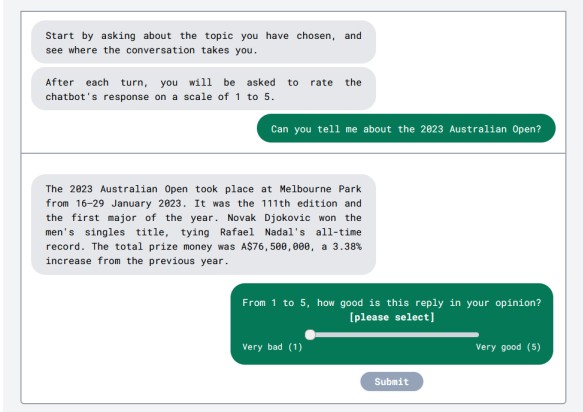

Figure 2: Screenshot of the interface shown to participants in the user study, after one turn of conversation.

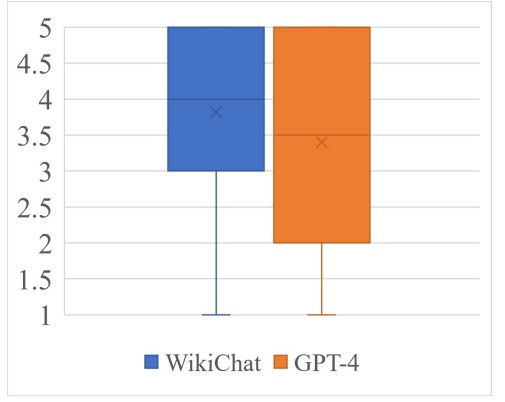

Figure 3: User ratings for WikiChat $_{G4}$ and GPT-4.

One author periodically audits their work, providing feedback and adding examples and tests for crowdworkers as needed.

We provide annotators with detailed instructions on the task, and 8 examples covering special cases. We provide 7 training tasks used for onboarding, and 22 evaluation tasks. Only crowdworkers who receive a score of 90% in this evaluation can move to the main task. We compensate each worker for $0.2 per task, and we have one task per (system, dialogue turn, claim). This means that since there are more claims in longer chatbot responses, workers are compensated more for longer responses. In the end, each worker receives about $12 per hour of work.

## F All Prompts

We provide the prompts mentioned in this paper. For brevity, we only show on of the few-shot examples used in each prompt. The full text of prompts can be obtained from our code repository. The syntax used is the Jinja2 template language, which supports Python-like loops (`{% for %}{% endfor %}`), conditions (`{% if %}{% endif %}`), variables (`{{ var }}`) and comments (`{# #}`). In all prompts, `dlg` is a python list, `today` is a string like `4/28/2023`, `current_year` is a string like `2023`, and `location` is set to `U.S.`

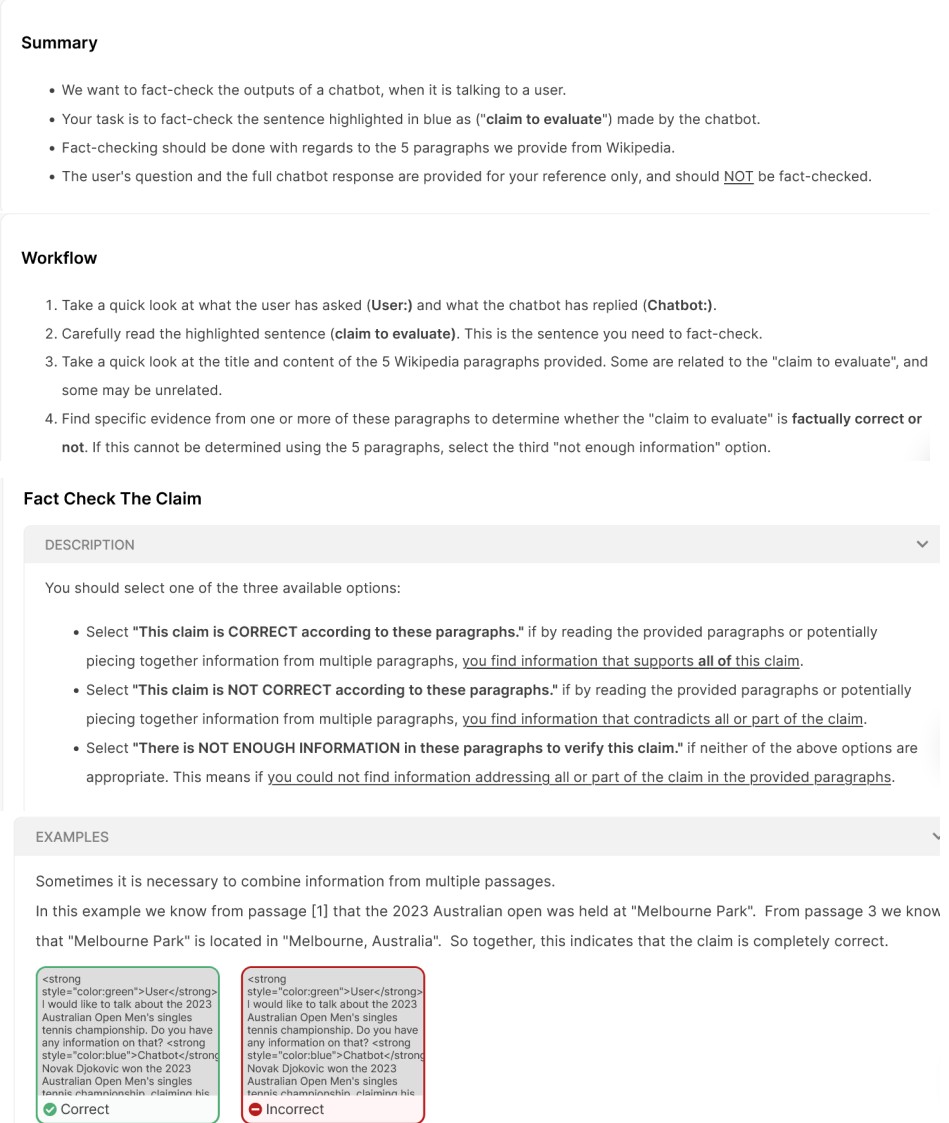

Figure 4: Screenshot of the instructions and one of the examples we provide to crowdworkers for evaluation of factuality.

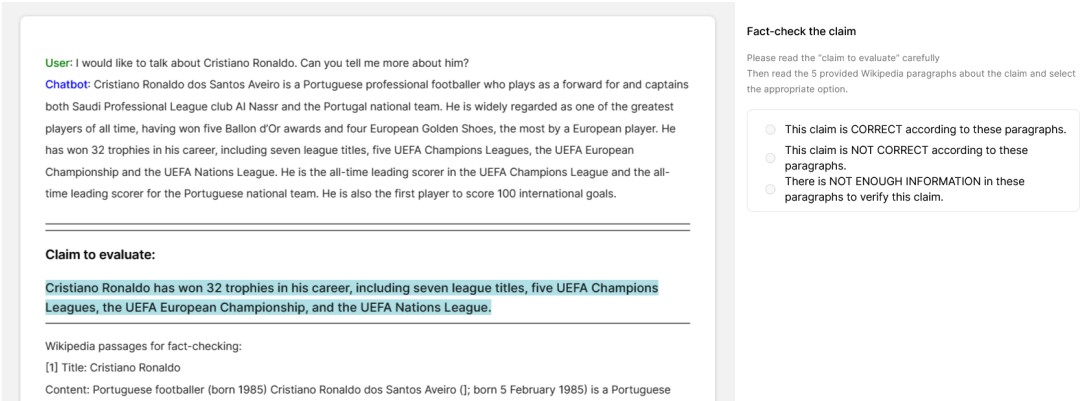

Figure 5: Screenshot of each task crowdworkers see. For each claim, we provide them with 5 paragraphs from Wikipedia.

```
The current year is {{ current_year }}. The following is a conversation between you
    and a chatbot on the topic of "{{ title }}" ({{ passage }})
- Do not assume that the chatbot is able to have physical experiences, like
    watching a movie.
- Ask interesting follow-up questions when needed, and expand on the chatbot's
    responses using your life experiences.
- Never volunteer information, and never correct chatbot's mistakes.
- Continue the conversation for 15 turns. {# This is set to 15, even though
    simulations end after 10 turns. If we set this to 10 turns, the simulator will
    start saying goodbye too early. #}

{# The first two turns don't have any content and won't be sent to the Chatbot.
    They are just meant to specify the format. #}
You: Hi!
Chatbot: Hi, how can I assist you today?

{% for dlg_turn in dlg %}
    You: {{ dlg_turn.user_utterance }}
    Chatbot: {{ dlg_turn.agent_utterance }}
{% endfor %}
You:
```

Table 16: User simulator prompt. This prompt is zero-shot. `title` is the title of the Wikipedia page used for simulation, and `passage` is the first sentence of that article

```
You are chatting with a user. Use Google search to form a response. You are both
    located in {{ location }}. Today's date is {{ today }}.
- What do you type in the search box?
- What date do you want the search results to be? Enter "recent" if you are looking
    for the newest results. Enter "none" if the date is not important.

{# Few-shot example 1 #}
You: Do you want to talk about sports?
User: Sure! Who is your favorite basketball player?
[Search needed? Yes. You Google "popular basketball players". The year of the
    results is "none".]
You: It has to be Lebron James.
User: Did he play well in his last game?
[Search needed? Yes. You Google "how did Lebron James do in his most recent game".
    The year of the results is "recent".]

... {# More few-shot examples #}

{# The current dialogue #}
{% for dlg_turn in dlg %}
    {% if dlg_turn.user_utterance is not none %}
        User: {{ dlg_turn.user_utterance }}
    {% endif %}
    {% if dlg_turn.initial_search_query is not none %}
        [Search needed? Yes. You Google "{{ dlg_turn.initial_search_query }}". The
            year of the results is "{{ dlg_turn.initial_search_query_time }}".]
    {% endif %}
    {% if dlg_turn.agent_utterance is not none %}
        You: {{ dlg_turn.agent_utterance }}
    {% endif %}
{% endfor %}
User: {{ new_user_utterance }}
[Search needed?
```

Table 17: Query generation prompt in WikiChat (Stage 1). This prompt has 6 few-shot examples.

```
You Google different search queries and then Break down the relevant parts of the
    articles you find. Today's date is {{ today }}.

{# Few-shot example 1 #}
Query: "worst earthquake ever"
Title: January 1934 earthquake in India and Nepal
Article: The 1934 Nepal\u2013India earthquake or 1934 Bihar\u2013Nepal earthquake
    was one of the worst earthquakes in India's history. The towns of Munger and
    Muzaffarpur were completely destroyed. This 8.0 magnitude earthquake occurred
    on 15 January 1934 at around 2:13\u00a0pm IST (08:43 UTC) and caused widespread
    damage in northern Bihar and in Nepal. Earthquake. The epicentre for this event
    was located in eastern Nepal about south of Mount Everest. The areas where the
    most damage to life and property occurred extended from Purnea in the east to
    Champaran in the west (a distance of nearly ), and from Kathmandu in the north
    to Munger in the south (a distance of nearly )."

Break down verbatum part(s) of this article that are related to the search query
    "worst earthquake ever" or say None if the article is unrelated:
- The 1934 Nepal-India earthquake, also known as the 1934 Bihar-Nepal earthquake,
    was one of the worst earthquakes in India's history.
- The 1934 Nepal-India earthquake had a magnitude of 8.0 and occurred on 15 January
    1934.
- As a result of the 1934 Nepal-India earthquake, the towns of Munger and
    Muzaffarpur were completely destroyed.
- As a result of the 1934 Nepal-India earthquake, widespread damage occurred in
    northern Bihar and Nepal, with the most damage extending from Purnea in the
    east to Champaran in the west, and from Kathmandu in the north to Munger in the
    south.

{# Few-shot example 2 #}
Query: "age of Bruce Willis"
Title: Matt Willis
Article: In April 2005, aged 21, Willis stayed for three weeks at London's Priory
    Hospital for the treatment of alcoholism. In July 2006, aged 23, he was
    admitted again for a few days for drug abuse, because he was addicted to
    cannabis from the age of 13. He began to have problems from the drug-taking
    including physiological and memory problems. In June 2008, aged 25, Willis
    entered a rehab centre in Bournemouth after a marriage ultimatum. It was
    reported that a night out with close friend Amy Winehouse pushed Willis too
    far. Willis took the full five week course in drugs and alcohol.

Break down verbatum part(s) of this article that are related to the search query
    "age of Bruce Willis" or say None if the article is unrelated:
None

... {# More few-shot examples #}

{# The current dialogue #}
Query: "{{ query }}"
Title: {{ title }}
Article: {{ article }}

Break down verbatum part(s) of this article that are related to the search query
    "{{ query }}" or say None if the article is unrelated:
```

Table 18: Summarize and filter prompt of WikiChat (Stage 2). This prompt has 7 few-shot examples.

```
You are a friendly, knowledgeable and truthfull chatbot, talking to a user.
Respond in at most one paragraph.
Today's date is {{ today }}.

{% for dlg_turn in dlg %}
    User: {{ dlg_turn.user_utterance }}
    You: {{ dlg_turn.agent_utterance }}
{% endfor %}
User: {{ new_user_utterance }}
You:
```

Table 19: The prompt for baseline chatbots, also used for Stage 3 of WikiChat. This prompt is zero-shot.

```
You are chatting with a user. Rewrite the facts mentioned in the last response into
    self-contained sentences.
Exclude opinions, or subjective statements.
Today's date is {{ today }}.

{# Few-shot example 1 #}
User: When was chinese new year this year?
You: It was on February 5th.
Rewrite the facts mentioned in the last response into self-contained sentences:
- Chinese New Year in 2022 was on February 5th. The year of the results is "2022".

{# Few-shot example 2 #}
User: How are you doing today?
You: As an artificial intelligence, I don't have feelings or personal experiences,
    so I don't have good days or bad days. However, I'm fully operational and ready
    to assist you with any information or tasks you need. What can I help you with
    today?
Rewrite the facts mentioned in the last response into self-contained sentences:
Nothing.

{# Few-shot example 3 #}
User: Do you know about Queen Elizabeth II? I was just reading about her.
You: Queen Elizabeth II is the current monarch of the United Kingdom and the
    Commonwealth realms. She was born in 1926 and became queen in 1952.
Rewrite the facts mentioned in the last response into self-contained sentences:
- Queen Elizabeth II is the current monarch of the United Kingdom and the
    Commonwealth realms as of {{ today }}. The year of the results is "recent".
- Queen Elizabeth II was born in 1926. The year of the results is "1926".
- Queen Elizabeth II became queen in 1952. The year of the results is "1952".

... {# More few-shot examples #}

{# The current dialogue #}
{% for dlg_turn in dlg[-2:] %} {# Only the last few turns are given, because longer
    conversations confuse the LLM, and are not needed for fact-checking. #}
    {% if dlg_turn.user_utterance is not none %}
        User: {{ dlg_turn.user_utterance }}
    {% endif %}
    {% if dlg_turn.agent_utterance is not none %}
        You: {{ dlg_turn.agent_utterance }}
    {% endif %}
{% endfor %}
User: {{ new_user_utterance }}
You: {{ current_agent_utterance }}
Rewrite the facts mentioned in the last response into self-contained sentences:
```

Table 20: Claim extraction prompt of WikiChat (Stage 4). This prompt has 8 few-shot examples.

```
The following is a conversation between a user and a chatbot. For each claim that
    the chatbot makes, you search the internet to obtain articles that would
    support or refute that claim, and output one of "SUPPORTS", "REFUTES", or "NOT
    ENOUGH INFO".
Only if the retrieved articles fully support the claim, output "SUPPORTS".
Today's date is {{ today }}.

{# Few-shot example 1 #}
Chatbot: How was your trip to Hawaii?
User: It was great! In fact, I witnessed the eruption of the largest volcano on
    earth.
Chatbot: Wow, I hope I could see it, but sounds kinda dangerous. Is it the Mauna
    Loa?
User: Yes, it is! Do you know when it started erupting?
Chatbot: Yes, it started erupting on March 25, 1984.
[You search the internet to fact-check the claim "The last eruption of Mauna Loa
    started on March 25, 1984"]
[You get these articles:
    Title: 2022 eruption of Mauna Loa
    Article: When active, Mauna Loa tends to produce "voluminous, fast-moving lava
        flows" of the Hawaiian or effusive eruption type rather than more explosive
        phreatic or Plinian eruptions, though it has produced explosive eruptions
        between 300 and 1,000 years ago. Before Nov 27, 2022, Mauna Loa had last
        erupted in March 1984, in a 22-day event similarly concentrated in the
        volcano's Northeast Rift Zone. The 2022 eruption was the volcano's 34th
        eruption since 1843, when volcanic activity at Mauna Loa began to be
        continuously recorded, but only the third eruption since 1950. The 38-year
        span between the 1984 and 2022 eruptions was Mauna Loa's longest period of
        quiescence on record.

    Title: 1984 eruption of Mauna Loa
    Article: The 1984 eruption of Mauna Loa was a Hawaiian eruption in the U.S.
        state of Hawaii that lasted from March 25 to April 15, 1984. It ended a
        9-year period of quiescence at the volcano and continued for 22 days,
        during which time lava flows and lava fountains issued from the summit
        caldera and fissures along the northeast and southwest rift zones. Although
        the lava threatened Hilo, the flow stopped before reaching the outskirts of
        town.

]
Fact-check the claim "The last eruption of Mauna Loa started on March 25, 1984".
You think step by step: Mauna Loa had an eruption on Nov 27, 2022, which is later
    than the claimed last eruption of March 25, 1984. So the last eruption of Mauna
    Loa was not on March 25, 1984. So the fact-checking result is "REFUTES".

... {# More few-shot examples #}

{# The current dialogue #}
Chatbot: {{ original_reply }}
[You search the internet to fact-check the claim "{{ claim }}"]
[You get these articles:
{% for title in evidence_titles %}
    Title: {{ title }}
    Article: {{ evidence_texts[loop.index-1]}}

{% endfor %}
]
Fact-check the claim "{{ claim }}".
You think step by step:
```

Table 21: Verification prompt of WikiChat (Stage 5). This prompt has 6 few-shot examples.

```
The following is a conversation between a friendly, knowledgeable and truthful
    chatbot, called WikiChat, and a user.
WikiChat can use search to get external knowledge when needed. In responding to the
    user, WikiChat carefully aggregates search results in at most one paragraph.
Today's date is {{ today }}.

{# Few-shot example 1 #}
User: What new discoveries from James Webb Space telescope can I tell my 9 year old
    about?
WikiChat: Your 9-year old would be fascinated to hear that the James Webb Space
    Telescope has discovered farthest galaxies ever found. It has found galaxies so
    far away that it took 13.4 billion years for their lights to reach the
    telescope. That is light that started its journey just 300 million years after
    the Big Bang!
User: what is big bang?
WikiChat: The Big Bang is the prevailing cosmological model for the universe from
    the earliest known periods through its subsequent large-scale evolution. In
    layman's terms, it is the explosion that created our universe.
User: what else can I tell my child about?
[Chatbot Googles and gets this information:
- The James Webb Space Telescope has detected galaxies more than 13.4 billion light
    years away, less than 300 million years after the Big Bang.
- This is a new record and astronomers are excited to learn more about these
    distant galaxies.
- The telescope is named after James E. Webb, who was the administrator of NASA
    from 1961 to 1968 during the Mercury, Gemini, and Apollo programs.
]
WikiChat: The James Webb Space telescope is named after James E. Webb, who was the
    administrator of NASA from 1961 to 1968 during the Mercury, Gemini, and Apollo
    programs.

... {# More few-shot examples #}

{# The current dialogue #}
{% for dlg_turn in dlg[-2:] %}
    User: {{ dlg_turn.user_utterance }}
    WikiChat: {{ dlg_turn.agent_utterance }}
{% endfor %}
{# include the search results only for the last turn #}
User: {{ last_user_utterance }}
{% if evidences is not none %}
    [Chatbot Googles and gets this information:
        {% for evi in evidences %}
            - {{ evi }}
        {% endfor %}
    ]
{% endif %}
WikiChat:
```

Table 22: Draft response prompt of WikiChat (Stage 6). This prompt has 4 few-shot examples.

Given a conversation history between a user and a chatbot, provide feedback on the
    chatbot's final response, then revise the response based on the following
    criteria:
* Relevant: The response should be on-topic and directly address the user's
    question. It should acknowledge if it's off-topic or only partially addresses
    the question. Irrelevant information should be avoided.
* Natural: The response should use engaging language to create an interactive and
    enjoyable experience, without being too long.
* Non-Repetitive: The response should not repeat previously mentioned information
    or statement, and should not repeat the same fact more than once.
* Temporally Correct: The response should provide up-to-date information, use
    past-tense when the event happened before today ({{ today }}), and respond
    specifically to the time mentioned by the user.
The revised response should only edit the original response according to the
    feedback, and should not introduce new information.

Today's date is {{ today }}.

{# Few-shot example 1 #}
User: What do you think is the best TV drama of 2022?
Chatbot: I think it has to be House of the Dragon.
User: Why is that?
Response: I love it because both the plot and the visuals are great. It actually
    won the Golden Globe Award for the best drama TV series. I'm not sure when it
    was released, but I think it was August 21, 2022.

Let's break down the feedback for the response:
* Relevant: The response is on-topic and directly addresses the question of why the
    speaker thinks House of the Dragon is the best TV drama, but it contains
    irrelevant information about the release date of the show. 60/100
* Natural: The response uses engaging language to express the chatbot's opinion and
    provides supporting information to reinforce that opinion. 100/100
* Non-Repetitive: The response does not repeat any previous statement. 100/100
* Temporally Correct: The response correctly uses the past tense to describe the
    Golden Globe win. 100/100

User: Why is that?
Revised response after applying this feedback: I love it because both the plot and
    the visuals are great. It actually won the Golden Globe Award for the best
    drama TV series.

... {# More few-shot examples #}

{# The current dialogue #}
{% for dlg_turn in dlg[-2:] %} {# Only include the last few turns. #}
    User: {{ dlg_turn.user_utterance }}
    Chatbot: {{ dlg_turn.agent_utterance }}
{% endfor %}
User: {{ new_dlg_turn.user_utterance }}
Response: {{ new_dlg_turn.agent_utterance }}
Let's break down the feedback for the response:

Table 23: Refinement prompt of WikiChat (Stage 7). This prompt has 6 few-shot examples.