# OpenReview forum: "WikiChat: Stopping the Hallucination of Large Language Model Chatbots by Few-Shot Grounding on Wikipedia"
_EMNLP/2023/Conference — EMNLP 2023 Findings_

### Official Review · Reviewer_zMxx · 2023-07-24

**Typos Grammar Style And Presentation Improvements:** See above.
**Soundness:** 2

**Excitement:**

3: Ambivalent: It has merits (e.g., it reports state-of-the-art results, the idea is nice), but there are key weaknesses (e.g., it describes incremental work), and it can significantly benefit from another round of revision. However, I won't object to accepting it if my co-reviewers champion it.

**Paper Topic And Main Contributions:**

The paper proposes a multi-stage pipeline that uses Wikipedia to refine LLM chatbot responses. Given a query, the pipeline involves passage retrieval and summarization (few-shot) from Wikipedia. Simultaneously, an original LLM response is generated, and claims are extracted (few-shot) from the response. Each claim is fact-checked with the retrieved passages (few-shot). A new response is drafted based on the claims and history (few-shot), and refined based on self-evaluated relevance, naturalness, non-repetitiveness, and temporal correctness (few-shot). This system (WikiChat) results in more factual responses while maintaining conversationality, based both on GPT-4-based evaluations and a user study. A smaller 7B LLaMA model can be fine-tuned on WikiChat responses (model distillation), outperforming larger models.

**Reasons To Accept:**

The paper describes a complex multi-part system that seems to be the result of many refinements. The proposed approach improves LLM factuality, which is a major problem for existing LLMs. The approach seems effective based on a user study and human-evaluated factuality.

**Reasons To Reject:**

Because WikiChat is the result of many iterative refinements, the specific contributions of each stage are not entirely clear. Although apparently still effective, some parts of the design may seem circular (e.g. using the same model for initial response generation, claim extraction, fact-verification, summarization, and response refinement).

The paper is quite difficult to follow. Implementation details are hard to find, because they are split between Section 3 (Design of WikiChat), Section 6 (Implementation), and the Appendix. For example, the information retrieval system description is split between Section 3.1.1 (l. 225-251) and Section 6 (l. 431-440), the model distillation approach is split between Section 4 (l. 305-328) and Section 6 (l. 418-430), etc. The methods for model distillation and evaluation somewhat interrupt the flow between Sections 3 and 6 (WikiChat design and implementation), and the evaluation/distillation results aren't presented until Sections 7 and 8. Because the design includes seven interdependent stages (each with substages), and there are multiple evaluation methods and experiments, this is quite confusing. The stage numbers in Figure 1 are not used throughout the rest of the paper.

While WikiChat seems effective, the paper might be trying to cover too many topics and implementations at once. There is the seven-stage system (with substages), which includes (among other things) information retrieval, summarization, fact-checking, model self-evaluation, and response refinement. There is an experiment with simulated conversations. There is an evaluation method with human annotators (factuality) and a model-based evaluation method (conversationality). There is a human study with additional human ratings (conversationality). There is also an experiment with model distillation.

**Reproducibility:**

2: Would be hard pressed to reproduce the results. The contribution depends on data that are simply not available outside the author's institution or consortium; not enough details are provided.

**Reviewer Confidence:**

4: Quite sure. I tried to check the important points carefully. It's unlikely, though conceivable, that I missed something that should affect my ratings.

---

> ### Author Rebuttal · Authors · 2023-08-29
>
> Thank you for your review and valuable comments.
>
> > ... the specific contributions of each stage are not entirely clear
>
> In section 7.3 and its multiple tables, we quantify the contributions of individual stages of our system. For instance:
>
> 1. We answer the question of “what would happen if we removed retrieval (stages 1, 2), or removed generation (stages 3, 4, 5)?”
> 1. We demonstrate and quantify the positive effect of fact-checking (stage 5), especially on tail and recent topics. Without this stage, about one third more unsupported claims from generation could have been included in the final response.
> 1. We show the improvements that refinement (stage 7) brings about.
>
> In addition, we try to motivate the reasoning behind including each stage throughout section 3.
>
> > … some parts of the design may seem circular (e.g. using the same model for initial response generation, claim extraction, fact-verification, summarization, and response refinement).
>
> Our approach is based on the principle of “avoiding premature optimization” by breaking down the task into as many subtasks as necessary, each of which is easy enough for GPT-X to solve, then optimize it by distilling down to a small model (Please see the discussion of latency below). Even though all stages use the same model, the instructions and few-shot examples given to them, make the same LLM behave very differently, in accordance with each of the 7 stages.
>
> When distilling, distilling all stages into the same LLaMA model has important benefits like the model sharing common skills (e.g. understanding multi-turn dialogues as inputs) across multiple tasks in our multi-task setting, and ease of deployment: we can deploy one model instead of 7, which would require significantly more GPU and system memory and disk space.
>
> > The paper is quite difficult to follow. Implementation details are hard to find, because they are split between Section 3 (Design of WikiChat), Section 6 (Implementation), and the Appendix.
> > ... the information retrieval system description is split between Section 3.1.1 (l. 225-251) and Section 6 (l. 431-440)
>
> > the model distillation approach is split between Section 4 (l. 305-328) and Section 6 (l. 418-430)
>
> Thank you for your feedback. Section 3 provides a high-level description and motivates the design, section 6 then provides the concrete "instantiation". For example, in section 3 we only use the term “LLM”, since any reasonably powerful LLM could be used for WikiChat; later in section 6 we mention the specific LLMs that we used in our experiments. Similarly for the information retrieval system, section 3.1.1 includes the design, independent of the specific modules used as the design does not depend on specifics of the underlying retrieval system.
> Regarding prompts, a description of “what” each stage and prompt is designed to achieve is included in the main text, but we moved generation hyperparameters and the text of our prompts to the appendix.
>
> In this way of presentation, we are following a lot of prior NLP papers, [1, 2, 3] just to name a few.
>
> > The stage numbers in Figure 1 are not used throughout the rest of the paper.
>
> Thank you for this suggestion, we will include these numbers in the text of the paper.
>
> > While WikiChat seems effective, the paper might be trying to cover too many topics and implementations at once. There is the seven-stage system (with substages), which includes (among other things) information retrieval, summarization, fact-checking, model self-evaluation, and response refinement. There is an experiment with simulated conversations. There is an evaluation method with human annotators (factuality) and a model-based evaluation method (conversationality). There is a human study with additional human ratings (conversationality). There is also an experiment with model distillation.
>
> At its core, the paper tries to answer one question: Can we effectively combat the hallucination problem of LLMs, using the one principle of breaking down the task into as many steps necessary, each of which simple enough that GPT-X could solve. Although it may look like we are solving many problems, they are just parts of a simple principle. To fully evaluate this principle, it is necessary to answer the questions raised: Are all these parts really needed? How does the system perform in the real world? Would our design add too much to the latency? Would the comparison with other systems depend on the topic of the conversation (e.g. head/tail/recent)? We answered all the questions in order to justify our methodology.
>
> With recent advances of LLMs, researchers are not held up by the long process of collecting training data anymore, and can instead focus on designing the most effective systems, not just the ones that are feasible to build given their budget or time constraints. As such, it is not surprising to see more papers that take on bigger tasks requiring solving several ML problems that in the past used to take one paper each to address.
>
> **Discussion of WikiChat’s Latency**
>
> It was an oversight not to have discussed the latency of the systems.
>
> WikiChat-G4 and WikiChat-G3.5 have the average end-to-end latency of ~25 and ~15 seconds per response, compared to ~2 seconds for the GPT-4 baseline. Retrieval from Wikipedia only accounts for ~200ms of this, and the rest is spent on waiting for LLM API calls. All API calls are done in parallel when possible, e.g. the stages 1-2 and 3-4 are independent and therefore parallelized.
>
> Distilling the WikiChat-G4 into WikiChat-LLaMA significantly lowers the latency. WikiChat-LLaMA has an average latency of ~4 seconds when measured on a single A100 GPU, and ~3 seconds when measured on a single VM with 4 A100 GPUs. This makes this system a viable alternative to the baselines: similarly low latency, while providing much more reliable information to the users.
>
> We would like to note that a naive implementation of WikiChat-LLaMA had ~30 seconds per response on a single GPU, and this +7X reduction in latency is the result of several speedup techniques: The latency improvements from most to least impactful are (1) fusing stages 6-7; (2) removing the chain-of-thought outputs (which are possible to do during distillation without any loss in quality, but would reduce system quality if done in the few-shot teacher WikiChat-G4); (3) using FlashAttention [4]; and other optimizations in the inference code.
>
>
> ### References
>
> [1] Chain of Thought Prompting Elicits Reasoning in Large Language Models, Wei et al, arXiv 2022
>
> [2] AutoQA: From Databases To QA Semantic Parsers With Only Synthetic Training Data, Xu et al, EMNLP 2020
>
> [3] ReAct: Synergizing Reasoning and Acting in Language Models, Yao et al, ICLR 2023
>
> [4] FlashAttention: Fast and Memory-Efficient Exact Attention with IO-Awareness, Dao et al, NeurIPS 2022

---

### Official Review · Reviewer_BJ2C · 2023-08-05

**Soundness:** 3

**Excitement:**

3: Ambivalent: It has merits (e.g., it reports state-of-the-art results, the idea is nice), but there are key weaknesses (e.g., it describes incremental work), and it can significantly benefit from another round of revision. However, I won't object to accepting it if my co-reviewers champion it.

**Missing References:**

Papers on continuous and lifelong learning in dialogue systems, especially knowledge acquisition ones, e.g.,
[Mazumder & Liu, arXiv2022] Lifelong and Continual Learning Dialogue Systems.
[Mazumder et al., SIGDIAL2019] Lifelong and Interactive Learning of Factual Knowledge in Dialogues.

Papers on knowledge-grounded dialogue focusing on knowledge selection e.g.,
[Li et al., NAACL2022] Enhanced Knowledge Selection for Grounded Dialogues via Document Semantic Graphs
[Sun et al., EACL2023] Generative Knowledge Selection for Knowledge-Grounded Dialogues (this is a recent paper so you don't have to cite but FYI)

Papers on the user simulator e.g.,
[Wan et al., EMNLP2022] A Unified Dialogue User Simulator for Few-shot Data Augmentation.



**Paper Topic And Main Contributions:**

This paper proposes a pipeline with LLMs for knowledge-grounded response generation utilizing Wikipedia. The proposed system has seven steps to generate a response, including a knowledge retriever querying Wikipedia or a fact-check module to verify the generated text. The authors further distilled LLaMA so that they can release the model publicly, and then incorporate it with the user simulator for evaluation. The experimental results suggest that the proposed approach is effectively building a dialogue system that can incorporate knowledge.

**Questions For The Authors:**

1. I'm not really sure that you can conclude that current LLMs are not making good chatbots because they cannot address something up-to-date (the first paragraph in the Introduction). This is more on the limitation of the current ML paradigm, and thus continual and lifelong learning is actively investigated.

2. What is the definition of conversationality in this paper? Are you claiming that knowledge-grounded dialogue is not conversational? (the second & third paragraphs in the Introduction)

3. What is the difference between a system that the Wizard of Wikipedia paper intended? This pipeline looks like an extended version of their system with the current state-of-the-art LLMs with access to Wikipedia. I do recognize that there are additional components such as summarize & filter or fact-check, but after all, these are just to ensure the performance.

4. How is the response time of WikiChat? It includes a lot of steps and I'm wondering whether the response generation can be done timely.

5. Have you conducted experiments to verify the quality of the user simulator?

6. What are head/tail/recent topics?

**Reasons To Accept:**

1. The paper introduces a strong baseline and will release a distilled model that can benefit the community. The paper can foster the research on knowledge-related research in dialogue systems, especially those with LLMs.

2. The user simulator part is interesting if elaborated more in detail.

**Reasons To Reject:**

1. Although seemingly effective, the proposed pipeline is rather engineering and weak in novelty. Except they have more staged modules, there is not much difference between conventional knowledge ground dialogue systems with a knowledge retriever & response generator.

2. The paper is well-organized and easy to follow. However, the tone/wording of the paper is weirdly non-scientific and bold. For each paragraph of the Introduction, there is a big claim that I cannot agree on without any proper grounding (citation, understandable background, etc.). It also feels like the paper tries to understate/trivialize the existing works in the field - to name a few, continual learning in dialogue systems, and knowledge-grounded dialogue.


**Reproducibility:**

3: Could reproduce the results with some difficulty. The settings of parameters are underspecified or subjectively determined; the training/evaluation data are not widely available.

**Reviewer Confidence:**

4: Quite sure. I tried to check the important points carefully. It's unlikely, though conceivable, that I missed something that should affect my ratings.

**Typos Grammar Style And Presentation Improvements:**

I strongly recommend rewriting the whole Introduction.

It may be better to change the numbering of the Tables in the Appendix (e.g., Table A.1 if it is Section A).

---

> ### Author Rebuttal · Authors · 2023-08-29
>
> > For each paragraph of the Introduction, there is a big claim that I cannot agree on without any proper grounding (citation, understandable background, etc.). It also feels like the paper tries to understate/trivialize the existing works in the field - to name a few, continual learning in dialogue systems, and knowledge-grounded dialogue.
>
> > I strongly recommend rewriting the whole Introduction.
>
> > Question 1: I'm not really sure that you can conclude that current LLMs are not making good chatbots because they cannot address something up-to-date (the first paragraph in the Introduction). This is more on the limitation of the current ML paradigm, and thus continual and lifelong learning is actively investigated.
>
> Thank you for your comment on the introduction - we will rewrite it, and include the citations you have kindly provided.
>
> Absolutely, there are several lines of work that tackle the hallucination problem. We focused a lot on the IR approaches (Lines 52-65) because there is a lot of related work in this area. More general external memory approaches like kNN-LLM [2], and continual learning as applied to LMs [3] also show great promise. We should have brought up these approaches early in the introduction; we will correct this in the final version.
>
> Our experiments show that the factuality of current LLMs is poor for not just recent information, but also for long tail entities and specific details, which are generally quite challenging to learn and memorize for LLMs. This means that WikiChat is complementary to approaches like continual learning. WikiChat is built on top of the strong generative power of the underlying LLM to generate facts in stage 3, so if the underlying LLM is more factual and knowledgeable as a result of continual learning, that will help WikiChat as well. In that case, WikiChat’s role would be covering more of the tail and minute details via information retrieval.
>
> > ... not much difference between conventional knowledge ground dialogue systems with a knowledge retriever & response generator
>
> > Question 3: What is the difference between a system that the Wizard of Wikipedia paper intended? This pipeline looks like an extended version of their system with the current state-of-the-art LLMs with access to Wikipedia. I do recognize that there are additional components such as summarize & filter or fact-check, but after all, these are just to ensure the performance.
>
> Thank you for your valuable comments. Our approach is based on the principle of "avoiding premature optimization" by breaking down the task into as many subtasks as necessary, each of which is easy enough for GPT-X to solve, then optimize it by distilling down to a small model  (Please see the discussion of latency below). As a result, our system has a 7-stage pipeline, as opposed to the conventional 2 stages. This design is discovered through extensive experimentation to avoid the various problems we have observed. We will include the following analysis in the final paper to emphasize the differences between our pipeline and conventional systems.
>
> We observe that conventional “retrieve-then-generate” dialogue systems have the following problems:
>
> 1. Existing retrieve-then-generate models produce ungrounded facts. When retrieval does not contain all the needed information, the generation stage is expected to pick up the slack, but unfortunately this produces ungrounded facts. In the Wizard of Wikipedia dataset, crowdworkers are free to not use retrieval results and make up their own response, which affects systems like Atlas that are trained with it. For BingChat, only 58.7% of the facts generated are grounded in retrieval when tested on popular academic datasets [1]. To take up the slack from retrieval, WikiChat also generates in stage 3, but we follow that with fact-checking to avoid hallucination. The final two stages (6 and 7) are prompted to not introduce any new facts.
>
> 1. Existing retrieve-then-generate models do not say “I don’t know”, when there is no information available, and would rather hallucinate. In WikiChat, If none of the retrieval results are relevant or factually accurate, they are filtered out during filtering (in stage 2) and fact-checking (stage 5) respectively. Then refinement (stage 7) responds with a contextual “I don’t know” (Section D.1).
>
> 1. When retrieval results contain irrelevant information, we observe that retrieve-then-generate models often confuse the different pieces of information or are distracted by it, like multiple people with the same last name, or the date when an event took place. We have included a summarization and filtering step (stage 2) to prevent this problem.
>
> Our system is very different from these conventional “retrieve-then-generate” systems, because it is able to reach near-perfect factual accuracy. (Note that if the system does not say anything, it can reach 100% factual accuracy, so on top of factuality, we also measure informationality as part of conversationality to ensure that the responses contain enough relevant information as well.). Our contribution is novel because it is the first time we have an LLM-based system that is essentially hallucination-free. This is an important attribute for systems to be trustworthy so they can be deployed in practice.
>
> In the following example from our simulated evaluation, both information retrieval (stages 1 and 2) and LLM generation (stages 3, 4 and 5) are used for information curation, with the major difference that everything is fact-checked to be grounded and not hallucinated.
> The first and last sentences are from LLM generation and the middle sentence is from information retrieval. The contributions of each part are quantified in Lines 554-555.
>
>     User: Do you think Sting’s birthplace has influenced his music?
>     WikiChat-G4: Yes, Sting has been heavily influenced by his birthplace of Newcastle upon Tyne, England. As a child, he was inspired by the Queen waving at him from a Rolls-Royce, which diverted him from the shipyard prospect towards a more glamorous life. He was also influenced by the music he heard in local nightclubs he visited such as Club A'Gogo.
>
> **Question 1:**
> Please see the discussion above.
>
> **Question 2:**
> Prior work on knowledge-grounded dialogue has measured metrics like humanness/naturalness and knowledgeable [4], similar to informationality which we use, but measured using a different methodology. Many papers [5] include discussions of repetitive chatbot responses, though not measured as a metric. We will add this connection to prior work in the final version of the paper. We carefully select five metrics that are highly desirable for knowledge-grounded chatbots, and collectively refer to them as conversationality metrics: relevant, informational, natural, non-repetitive, and temporally correct (Lines 384- 394).
>
> We should have qualified that our claim “... lacking in the previous knowledge-grounded chatbots” (Line 50-51) refers to our findings when testing Atlas, a state-of-the-art knowledge-grounded chatbot, in Table 1. Conversations like the one in Table 6 (Appendix A) were quite common when we evaluated Atlas: often short responses that might ignore part of the user’s interest/question, and dry language that does not encourage further discussion.
>
> **Question 3:**
> Please see the discussion above.
>
> **Question 4:**
> It was an oversight not to have discussed the latency of the systems.
>
> WikiChat-G4 and WikiChat-G3.5 have the average end-to-end latency of ~25 and ~15 seconds per response, compared to ~2 seconds for the GPT-4 baseline. Retrieval from Wikipedia only accounts for ~200ms of this, and the rest is spent on waiting for LLM API calls. All API calls are done in parallel when possible, e.g. the stages 1-2 and 3-4 are independent and therefore parallelized.
>
> Distilling the WikiChat-G4 into WikiChat-LLaMA significantly lowers the latency. WikiChat-LLaMA has an average latency of ~4 seconds when measured on a single A100 GPU, and ~3 seconds when measured on a single VM with 4 A100 GPUs. This makes this system a viable alternative to the baselines: similarly low latency, while providing much more reliable information to the users.
>
> We would like to note that a naive implementation of WikiChat-LLaMA had ~30 seconds per response on a single GPU, and this +7X reduction in latency is the result of several speedup techniques: The latency improvements from most to least impactful are (1) fusing stages 6-7; (2) removing the chain-of-thought outputs (which are possible to do during distillation without any loss in quality, but would reduce system quality if done in the few-shot teacher WikiChat-G4); (3) using FlashAttention [6]; and other optimizations in the inference code.
>
> **Question 5:**
> Like other parts of our system, the user simulator has also gone through many rounds of improvements, in order to make the user closer to how users behave, as categorized by collecting a small set of dialogues we internally collected for development purposes. For instance, the way the simulator asks follow-up questions, how it handles chatbots' inconsistent responses etc.
>
> The main validation of our user simulator is the user study reported in Section 8. We show that for dialogues on recent topics, the results correlate with the simulator. As we discussed in the paper, the simulator allows us to evaluate dialogues involving tail topics, as it is difficult to find users to discuss tail topics.
>
> **Question 6:**
> Head, tail and recent topics are selected as follows:
>
> - Head: We sort all wikipedia articles from most viewed to least viewed, based on the total view count from the date that the article was published until the end of 2020. We stop at 2020 so that this subset is disjoint from the “recent” subset. Example topics include Wikipedia articles titled “Sting (musician)”, “Barack Obama”, and “Gmail”. The view count ranges from 68M to 16M for the head topics.
>
> - Tail: Reverse order of the head subset. Example topics include articles titled “Amelia Gething”, “Last Tycoon Stakes”, “2008 CONCACAF Women's Olympic Qualifying Tournament”. The view count for the tail subset is under 1000.
>
> - Recent: As explained in Line 1002, we select the most edited Wikipedia articles in the first four months of 2023. Example topics include articles like “2023 Speaker of the United States House of Representatives election” and “Yeti Airlines Flight 691”, which happened to be big news stories of 2023.
> For each subset, we select from the top of the sorted list, ignoring topics that involve violent or explicit content (Line 468), until we have selected 20 topics in that subset. View count and edit information are obtained using the Wikimedia API.
>
> The main motivation for these subsets is to quantify the gap in factual accuracy of LLMs when information about a topic is scarce in their pre-training data (tail) or absent (recent). As mentioned in Line 460, Wikipedia view count is simply a proxy for this, given that the pre-training data for GPT-3.5 and GPT-4 is not publicly available. For the same purpose, the “recent” subset is chosen from 2023 to make sure it is not included in the pre-training data of these models.
> We will include this more detailed version of how topics are chosen in the final version of the paper.
>
> ### References
> [1] Evaluating Verifiability in Generative Search Engines, Liu et al, arXiv 2023
>
> [2] Generalization through Memorization: Nearest Neighbor Language Models, Khandelwal et al, ICLR 2020
>
> [3] TemporalWiki: A Lifelong Benchmark for Training and Evaluating Ever-Evolving Language Models, Jang et al, EMNLP 2022
>
> [4] ACUTE-EVAL: Improved dialogue evaluation with optimized questions and multi-turn comparisons, Li et al, arXiv 2019
>
> [5] Recipes for building an open-domain chatbot, Roller et al, EACL 2021
>
> [6] FlashAttention: Fast and Memory-Efficient Exact Attention with IO-Awareness, Dao et al, NeurIPS 2022

---

### Official Review · Reviewer_ApEL · 2023-08-05

**Soundness:** 3

**Excitement:**

4: Strong: This paper deepens the understanding of some phenomenon or lowers the barriers to an existing research direction.

**Paper Topic And Main Contributions:**

This paper presents WikiChat, an innovative chatbot that harnesses both retrieved data from Wikipedia and generative content from LLMs (Large Language Models). The proposed 7-step pipeline enables WikiChat to produce factual conversation content with minimal hallucinations, while also being capable of discussing up-to-date topics beyond its pre-training data. Additionally, the study demonstrates the successful distillation of WikiChat into a smaller LLaMA model, exhibiting comparable performance. Through extensive experiments and automatic and human evaluations on GPT 3.5, GPT 4.0, and LLaMA, the effectiveness of this method is established concerning factuality and conversationality.

**Questions For The Authors:**

Question A: Line 1037 states, "In order to verify that using GPT-4 for conversationality metrics (Section 5) is indeed reasonable, we compare its scores against two authors of this paper." Could this potentially introduce bias as the authors themselves are involved in this verification process?

Question B: I am not clear how they split the claim into multiple simple ones, though I have checked Table 19 Claim splitting prompt (truncated) of WikiChat. What is the specific prompt for splitting? Do you have the accuracy of this split?

Question C: Line 115 states “hallucinating 27 times less than GPT-4”, what is the total number of topics and factual claims?

Question D: Line 464. w\What is the distribution of the head, tail, and recent topics? What is the number of views threshold of them?

Question E: How is N_IR = 3 and N_evidence = 2 determined?


**Reasons To Accept:**

S1 The proposed method looks sound and achieved state-of-the-art performance. I genuinely appreciate the tremendous efforts invested in this research.

S2 The effectiveness of the method was evaluated through comprehensive experiments and evaluations.

S3 The paper commits to releasing its code and model upon acceptance, thereby presenting a valuable resource for the natural language generation community.


**Reasons To Reject:**

W1: Certain method/experiment/evaluation details lack clarity, as indicated in Questions.

W2: To enhance the paper, it is recommended to include an evaluation of the system's latency, given the presence of 7 stages involved in generating the response, with information retrieval and fact-checking potentially requiring additional processing time.


**Reproducibility:**

3: Could reproduce the results with some difficulty. The settings of parameters are underspecified or subjectively determined; the training/evaluation data are not widely available.

**Reviewer Confidence:**

4: Quite sure. I tried to check the important points carefully. It's unlikely, though conceivable, that I missed something that should affect my ratings.

**Typos Grammar Style And Presentation Improvements:**

P1 See Questions.

P2 Line 421---423 mentions head, tail, and recent topics before explaining what they are in Lines 463—465.

---

> ### Author Rebuttal · Authors · 2023-08-29
>
> Thank you for your review and valuable comments. We will incorporate the information below to the final version of the paper.
>
> **Discussion of WikiChat’s Latency**
>
> It was an oversight not to have discussed the latency of the systems.
>
> WikiChat-G4 and WikiChat-G3.5 have the average end-to-end latency of ~25 and ~15 seconds per response, compared to ~2 seconds for the GPT-4 baseline. Retrieval from Wikipedia only accounts for ~200ms of this, and the rest is spent on waiting for LLM API calls. All API calls are done in parallel when possible, e.g. the stages 1-2 and 3-4 are independent and therefore parallelized.
>
> Distilling the WikiChat-G4 into WikiChat-LLaMA significantly lowers the latency. WikiChat-LLaMA has an average latency of ~4 seconds when measured on a single A100 GPU, and ~3 seconds when measured on a single VM with 4 A100 GPUs. This makes this system a viable alternative to the baselines: similarly low latency, while providing much more reliable information to the users.
>
> We would like to note that a naive implementation of WikiChat-LLaMA had ~30 seconds per response on a single GPU, and this +7X reduction in latency is the result of several speedup techniques: The latency improvements from most to least impactful are (1) fusing stages 6-7; (2) removing the chain-of-thought outputs (which are possible to do during distillation without any loss in quality, but would reduce system quality if done in the few-shot teacher WikiChat-G4); (3) using FlashAttention [1]; and other optimizations in the inference code.
>
> **Question A:** We use GPT-4 as an evaluator of conversationality primarily because of the success of  such an approach reported in papers cited in Lines 395-396; these papers measure GPT-4’s ratings against crowdworkers. Our comparison against two authors is an additional sanity check. Our exact claim would be that for these soft conversationality metrics, the difference between GPT-4 and one author is not more than the difference between two authors.
>
> **Question B:** Table 19 contains the prompt for claim splitting; we include only one example, omitting 7 others for the sake of brevity. The output is a list of sentences, each of which is referred to as a “claim”. There are several criteria we would like the outputs of this prompt to have, as we partially described in Line 258:
>
> 1. Each output claim needs to be as atomic as possible, meaning small enough that it can be fully supported or refuted, without the possibility that part of it is supported and others not.
>
> 1. Each claim needs to be self-contained, without references to the dialogue history or other context. This can be implicit and therefore tricky for LLMs. For example, consider the statement “It's hard to pick just one, but some of the most popular TV dramas of last year were The Crown and The Queen's Gambit.” Claim splitting therefore
>
>     a. resolves co-references
>
>     b. resolves context-dependent time information like “current” and “last year”
>
>     The output then is the two self-contained claims “The Crown is one of the most popular TV dramas of 2022.”, “The Queen’s Gambit is one of the most popular TV dramas of 2022.”.
>
> 1. Parts of each input statement that are subjective opinions (e.g. “Absolutely! He [Hayao Miyazaki] is just incredible”) or otherwise do not contain objective facts (e.g. “I can help find that information for you”) should be ignored. Note that these cases happen because claim splitting is applied to the output of an LLM prompted to be a chatbot (in stage 3), which can contain chit-chat if that is what fits the conversation.
>
> Through many rounds of trial and error, we realized that instructing LLMs to act as a fact-checker by completing the sentence “To fact-check only your last response, you Google …” gives better results than directly instructing them to split claims given the above criteria. 8 few-shot examples help address edge cases and clarify the criteria.
>
> We manually inspected the outputs on a dev set after each round we made a change to improve the results, and the accuracy of this split in our final prompt for GPT-4 is 97%. Meaning that out of the splitted claims, about 97% match our criteria. This very high accuracy is why it makes sense that in Section 5 (Line 358) we use the claim splitting prompt with GPT-4 in order to assist with the human evaluation of factuality as well.
>
> **Question C:** Our user study has 40 participants, and each person talks to one of the two chatbots for 5 turns (Line 599). So each of the two systems under study generates 100 turns. The GPT-4 baseline responses contain 163 claims overall, 42.9% of which (70 claims) are factually correct. WikiChat-G4 responses contain 239 claims, 97.9% of which (234 claims) are factually correct.
>
> **Question D:** Head, tail and recent topics are selected as follows:
>
> - Head: We sort all wikipedia articles from most viewed to least viewed, based on the total view count from the date that the article was published until the end of 2020. We stop at 2020 so that this subset is disjoint from the “recent” subset. Example topics include Wikipedia articles titled “Sting (musician)”, “Barack Obama”, and “Gmail”. The view count ranges from 68M to 16M for the head topics.
>
> - Tail: Reverse order of the head subset. Example topics include articles titled “Amelia Gething”, “Last Tycoon Stakes”, “2008 CONCACAF Women's Olympic Qualifying Tournament”. The view count for the tail subset is under 1000.
>
> - Recent: As explained in Line 1002, we select the most edited Wikipedia articles in the first four months of 2023. Example topics include articles like “2023 Speaker of the United States House of Representatives election” and “Yeti Airlines Flight 691”, which happened to be big news stories of 2023.
> For each subset, we select from the top of the sorted list, ignoring topics that involve violent or explicit content (Line 468), until we have selected 20 topics in that subset. View count and edit information are obtained using the Wikimedia API.
>
> The main motivation for these subsets is to quantify the gap in factual accuracy of LLMs when information about a topic is scarce in their pre-training data (tail) or absent (recent). As mentioned in Line 460, Wikipedia view count is simply a proxy for this, given that the pre-training data for GPT-3.5 and GPT-4 is not publicly available. For the same purpose, the “recent” subset is chosen from 2023 to make sure it is not included in the pre-training data of these models.
> We will include this more detailed version of how topics are chosen in the final version of the paper.
>
> **Question E:** These values were selected after looking at the top-N recall of our retriever using the following method: We informally estimate the top-N “recall” (i.e. what is the probability that top-N documents contain relevant information to the query) of the ColBERT retriever on a sample set of queries that were generated from stage 1 of our pipeline in an early test. Note that due to the differences in the distribution of queries, this value is different from reported numbers on other datasets in the ColBERT paper, and we are using a relaxed definition or “recall” here, where each query can potentially have many gold documents.
>
> We estimate this value using the cumulative probability that the first N documents are relevant to the query. We obtain these probabilities by taking the softmax of the retrieval similarity scores over a large number of retrieved documents. The top-5 recall was more than 0.9, and manually inspecting top-5 results showed that in most cases the first three results contain more than enough information to satisfy the query. Setting N_evidence=3 in practice led to LLMs making more mistakes during fact-checking, likely due to the fact that the input is long and contains a lot of distracting information. We decided to reduce this to N_evidence=2 to mitigate this issue. In general, lowering N_evidence can at worst increase the number of claims that should be classified as “supported” or “refuted” but are misclassified as “not enough information”. This does not negatively impact factual accuracy, since we do not include “not enough information” claims in the final response (Line 275).
>
>
> ### Reference
> [1] FlashAttention: Fast and Memory-Efficient Exact Attention with IO-Awareness, Dao et al, NeurIPS 2022

---

### Meta-Review · Area_Chair_BhZR · 2023-09-18

**Recommendation:** 3

**Metareview:**

Based on the reviews, the paper presents a multi-stage pipeline called WikiChat that combines retrieved data from Wikipedia with generative content from LLMs to produce factual and up-to-date conversation content. The reviewers generally agree on the soundness of the proposed method, its state-of-the-art performance, and the value of releasing the code and model to the community. However, there are some concerns and suggestions for improvement.

Review 1 appreciates the effort invested in the research and finds the proposed method sound. The author suggests clarifying certain method/experiment/evaluation details and including an evaluation of system latency. Review 2 sees the proposed pipeline as rather engineering and lacking novelty, criticizing the non-scientific tone and bold claims in the introduction. They also make several questions about the definition of conversationality, comparison with the Wizard of Wikipedia system, response time, and quality of the user simulator. Additionally, they recommend rewriting the whole introduction. Review 3 acknowledges the complex nature of the pipeline and its effectiveness in improving factuality. However, they find it difficult to follow due to the scattered implementation details and the paper covering too many topics at once.

Overall, the reviewers agree that the proposed method improves factuality and conversationality, and achieves state-of-the-art performance. However, improvements are needed in terms of clarity, organization, and presentation. The reviewers raise concerns about clarity in method/experiment/evaluation details, the definition of conversationality, and the comparison with existing systems. They also suggest including an evaluation of system latency and verifying the quality of the user simulator. The paper's presentation, including the introduction and organization of details, should be revised for better readability.

In terms of scores, the reviewers generally rate the soundness of the paper as good (3) and the excitement as ambivalent (3). While the paper has merits such as reporting state-of-the-art results, there are also weaknesses that may require further revision. However, no reviewer strongly objects to accepting the paper.

In summary, the paper presents an innovative chatbot pipeline that combines retrieved data and generative content for factual and up-to-date conversation. While the method shows promise and achieves state-of-the-art performance, improvements are needed in terms of clarity, organization, and presentation. The paper covers multiple topics and implementations, making it challenging to follow. However, with revisions addressing the reviewers' concerns, the paper can be much better.

---

### Meta-Review · Senior_Area_Chairs · 2023-10-05

**Recommendation:** 3

**Metareview:**

meta review

---

### Decision · Program_Chairs · 2023-10-07

**Decision:**

Accept-Findings

**Comment:**

Based on the reviews, the paper presents a multi-stage pipeline called WikiChat that combines retrieved data from Wikipedia with generative content from LLMs to produce factual and up-to-date conversation content. The reviewers generally agree on the soundness of the proposed method, its state-of-the-art performance, and the value of releasing the code and model to the community. However, there are some concerns and suggestions for improvement.

Review 1 appreciates the effort invested in the research and finds the proposed method sound. The author suggests clarifying certain method/experiment/evaluation details and including an evaluation of system latency. Review 2 sees the proposed pipeline as rather engineering and lacking novelty, criticizing the non-scientific tone and bold claims in the introduction. They also make several questions about the definition of conversationality, comparison with the Wizard of Wikipedia system, response time, and quality of the user simulator. Additionally, they recommend rewriting the whole introduction. Review 3 acknowledges the complex nature of the pipeline and its effectiveness in improving factuality. However, they find it difficult to follow due to the scattered implementation details and the paper covering too many topics at once.

Overall, the reviewers agree that the proposed method improves factuality and conversationality, and achieves state-of-the-art performance. However, improvements are needed in terms of clarity, organization, and presentation. The reviewers raise concerns about clarity in method/experiment/evaluation details, the definition of conversationality, and the comparison with existing systems. They also suggest including an evaluation of system latency and verifying the quality of the user simulator. The paper's presentation, including the introduction and organization of details, should be revised for better readability.

In terms of scores, the reviewers generally rate the soundness of the paper as good (3) and the excitement as ambivalent (3). While the paper has merits such as reporting state-of-the-art results, there are also weaknesses that may require further revision. However, no reviewer strongly objects to accepting the paper.

In summary, the paper presents an innovative chatbot pipeline that combines retrieved data and generative content for factual and up-to-date conversation. While the method shows promise and achieves state-of-the-art performance, improvements are needed in terms of clarity, organization, and presentation. The paper covers multiple topics and implementations, making it challenging to follow. However, with revisions addressing the reviewers' concerns, the paper can be much better.|meta review